*Resource*

# A comparative analysis of the mobility of 45 proteins in the synaptic bouton

Sofiia Reshetniak[1,2] (iD), Jan-Eike Ußling[1], Eleonora Perego[3], Burkhard Rammner[1], Thomas Schikorski[4], Eugenio F Fornasiero[1] (iD), Sven Truckenbrodt[1,2], Sarah Köster[3,5] & Silvio O Rizzoli[1,5,*] (iD)

## Abstract

**Many proteins involved in synaptic transmission are well known, and their features, as their abundance or spatial distribution, have been analyzed in systematic studies. This has not been the case, however, for their mobility. To solve this, we analyzed the motion of 45 GFP-tagged synaptic proteins expressed in cultured hippocampal neurons, using fluorescence recovery after photobleaching, particle tracking, and modeling. We compared synaptic vesicle proteins, endo- and exocytosis cofactors, cytoskeleton components, and trafficking proteins. We found that movement was influenced by the protein association with synaptic vesicles, especially for membrane proteins. Surprisingly, protein mobility also correlated significantly with parameters as the protein lifetimes, or the nucleotide composition of their mRNAs. We then analyzed protein movement thoroughly, taking into account the spatial characteristics of the system. This resulted in a first visualization of overall protein motion in the synapse, which should enable future modeling studies of synaptic physiology.**

**Keywords** diffusion; movement; protein mobility; synapse; vesicle
**Subject Category** Neuroscience
**The EMBO Journal (2020) 39: e104596**

## Introduction

Synaptic transmission is one of the best-known cellular pathways, with most of its components being thoroughly annotated in functional terms (Koopmans *et al*, 2019). Within the synapse, the synaptic vesicle recycling pathway has been analyzed in very high detail, for several decades. This pathway involves the fusion of synaptic vesicles at the active zone (exocytosis), which is followed by the retrieval of the fused vesicle molecules (endocytosis), and by the reformation of new fusion-competent vesicles (Sudhof, 2004; Haucke *et al*, 2011; Rizzoli, 2014). The copy

numbers of the molecules involved in synaptic vesicle recycling are known relatively well (Takamori *et al*, 2006; Wilhelm *et al*, 2014). Many other features of these proteins have also been analyzed in systematic studies, ranging from their overall spatial distributions (Wilhelm *et al*, 2014) to their translation in relation to synaptic function (Schanzenbächer *et al*, 2016) or to their lifetimes, both *in vitro* (Dörrbaum *et al*, 2018) and *in vivo* (Fornasiero *et al*, 2018). Such systematic studies have revealed numerous unexpected features, including strong correlations between protein functions and their lifetimes (Dörrbaum *et al*, 2018), or links between the protein and mRNA structures and a many functional parameters such as the translation rates (Mandad *et al*, 2018).

However, one important characteristic of synaptic proteins, their mobility, has not been the subject of large systematic studies. The movement of synaptic organelles, and especially of synaptic vesicles, has been thoroughly investigated (Rothman *et al*, 2016). Active transport of molecules to and from synapses has also been measured in numerous studies (Hirokawa *et al*, 2010; Roy, 2014). The movement of individual proteins in synapses has been less investigated, in studies that typically only targeted one or a handful of presynaptic molecules (e.g., Kamin *et al*, 2010; Ribrault *et al*, 2011; Albrecht *et al*, 2016). Such studies resulted in valuable insights for the respective proteins, but did not enable further analyses of, for example, protein structure in relation to synaptic mobility. Many important questions could only be approached by systematic works targeting multiple proteins simultaneously. For example, is the synaptic protein mobility determined by their size, or is their movement dominated by specific interactions with other synaptic components, rendering size effects irrelevant? As another example, several biochemical and imaging experiments have demonstrated thoroughly that the vesicle cluster binds to substantial amounts of cofactor proteins (Shupliakov, 2009; Denker *et al*, 2011a; Fornasiero *et al*, 2012; Milovanovic & Camilli, 2017). How does this relate to the protein movement? Is this effect relevant for both soluble and membrane proteins? At the same time, many functional protein parameters are known to depend on the respective protein and mRNA sequences, as mentioned above (Mandad *et al*,

1 Institute for Neuro- and Sensory Physiology and Biostructural Imaging of Neurodegeneration (BIN) Center, University Medical Center Göttingen, Göttingen, Germany
2 International Max Planck Research School for Molecular Biology, Göttingen, Germany
3 Institute for X-Ray Physics, University of Göttingen, Göttingen, Germany
4 Department of Neuroscience, Universidad Central del Caribe, Bayamon, PR, USA
5 Cluster of Excellence "Multiscale Bioimaging: from Molecular Machines to Networks of Excitable Cells" (MBExC), University of Göttingen, Göttingen, Germany
*Corresponding author. Tel: +49551395911; E-mail: srizzol@gwdg.de

2018). Could one determine such correlations also for protein movement parameters?

Such questions are difficult to explore in the absence of a large protein movement dataset. To address this challenge, we aimed to measure the mobility of multiple proteins in the synaptic bouton and in the axon. We obtained measurements for 47 proteins, including controls such as free cytosolic GFP, or membrane-bound GFP. We relied on the overexpression of GFP-tagged variants of the proteins, which is the only efficient solution when large numbers of constructs need to be analyzed. To minimize, as much as possible, the deleterious effects of GFP fusion and overexpression, we only used GFP chimeras that had been validated in the past, and we made efforts to only investigate neurons with mild expression levels. We found that the average overexpression levels ranged from ~1.2-fold to 2-fold, compared to the normal expression levels, for multiple tested proteins (albeit overexpression could reach higher values in individual neurons and synapses). We also controlled for possible connections between overexpression levels and protein mobility behaviors, and found no substantial correlations for any of the analyzed proteins. Finally, the motion measurements we obtained could reproduce well several similar measurements of (i) fluorophore-tagged native proteins and vesicles; (ii) GFP-tagged proteins expressed in mice after knock-in procedures. Overall, this suggests that our measurements reproduce well the behavior of the native proteins.

Having thus obtained a large dataset of comparative movement measurements for synaptic proteins, we proceeded to solve the questions posed above. Our results demonstrated that, for example, protein size has a very limited effect on synaptic mobility and that protein movement parameters correlate to many other cell biology parameters. We then analyzed the movement data by a modeling approach, based on the structural features of the synapses. This resulted in movement rate estimates (diffusion coefficients) for the different proteins in the axon, in the synapse, and in the vesicle cluster. These movement rates (and/or similar movement rates obtained by more complex models, which can be readily performed using our data) will be employed in the future in investigating the molecular kinetics of synaptic function (*e.g.,* exo- or endocytosis) with higher precision than currently possible.

## Results

### An overview of the proteins analyzed

The mobility of membrane proteins has been analyzed by quantum dot tracking in the past (e.g., Ribrault *et al*, 2011; Albrecht *et al*, 2016). As this is not a feasible labeling option for cytosolic proteins, and as its use for tracking membrane proteins in synapses has also been recently criticized (Lee *et al*, 2017), we decided to pursue this study mainly by fluorescence recovery after photobleaching (FRAP) (Axelrod *et al*, 1976). Commercial quantum dots have a relatively large size (~20 nm in diameter) and are typically coupled to their targets using antibodies (~10–15 nm in diameter). This renders the labels substantially larger than their targets, which may influence the target movement. Moreover, such labels may be unable to penetrate in areas as the synaptic cleft (Lee *et al*, 2017). GFP, with a diameter of 2–3 nm, is substantially smaller than even low-size,

non-commercial quantum dots (~5–10 nm). Moreover, GFP does not require bridging molecules, as antibodies, for linking to the target protein. Therefore, GFP is expected to affect the protein behavior to a substantially lower extent than the quantum dots. We thus expressed 45 different proteins tagged with monomeric enhanced GFP (mEGFP) in mature hippocampal cultured neurons, focusing on proteins known to participate either in exo- or in endocytosis. We employed proteins whose tagging has been tested and validated in the past in various assays (Fig 1, Table EV1). All of the tagged proteins we employed have been demonstrated to localize in the expected areas, and many have been used to rescue function in cells or animals lacking the wild-type protein (Fig 1, Table EV1). We have also analyzed how proteins were differentially distributed in the synapse and in the axon, both for the tagged proteins (measuring the mEGFP fluorescence in the two compartments) and for the same untagged endogenous proteins (relying on immunostainings; Appendix Figs S1 and S2). The measurements obtained with tagged or untagged proteins correlate well, suggesting that the presence of the mEGFP moiety does not induce major effects on protein localization. Overall, we analyzed proteins involved in exo- and endocytosis, along with *bona fide* synaptic vesicle proteins, endosomal proteins, cytoskeletal components, and different trafficking proteins located both in the cytosol and in the plasma membrane (Fig 1).

### The basic results: FRAP recovery rates and immobile fractions for the different proteins

Tagged proteins typically localized both to synaptic boutons and to the axonal compartment (Fig 2A and B). This enabled us to bleach both synaptic and axonal areas in live neurons, and to monitor the FRAP behavior of the proteins (Fig 2B) for both compartments. Fitting FRAP recovery curves with exponential rise to maximum equations (Fig 2C) provided recovery time constants ($\tau$) and immobile fractions in both axons and synapses (Fig 1D–F).

These values are presented in Table EV2 and are also shown in full detail in the large Appendix Fig S3. We used neurons that were allowed to behave normally, and to fire bursts of action potentials freely (at about 0.1 Hz, Truckenbrodt *et al*, 2018). This implies that the protein motion behavior we observed conforms to conditions of mild activity, which should involve, for example, some level of release of soluble proteins from the vesicle cluster [driven by rises in the $Ca^{2+}$ concentration and by the phosphorylation of key molecules such as synapsin (Cesca *et al*, 2010; Rizzoli, 2014; Milovanovic & Camilli, 2017)]. Heavy stimulation or activity inhibition may provide different results, but the results of such experiments would not be physiologically relevant (Denker *et al*, 2011b).

Since high expression levels can affect protein mobility (e.g., via saturation of binding sites on the cofactors of the respective proteins), we only analyzed cells with moderate expression of tagged proteins, as shown in Fig 3A–C.

Additionally, to evaluate a potential correlation between the expression levels and protein mobility, we compared the protein abundance and the time constants obtained for each individual synapse or axonal region we analyzed (Appendix Fig S5). We found no significant correlation for any of the proteins. This suggests that the mobility rates we measured are not drastically affected by the protein concentration changes produced by the expression (within the range caused by overexpression in our experiments).

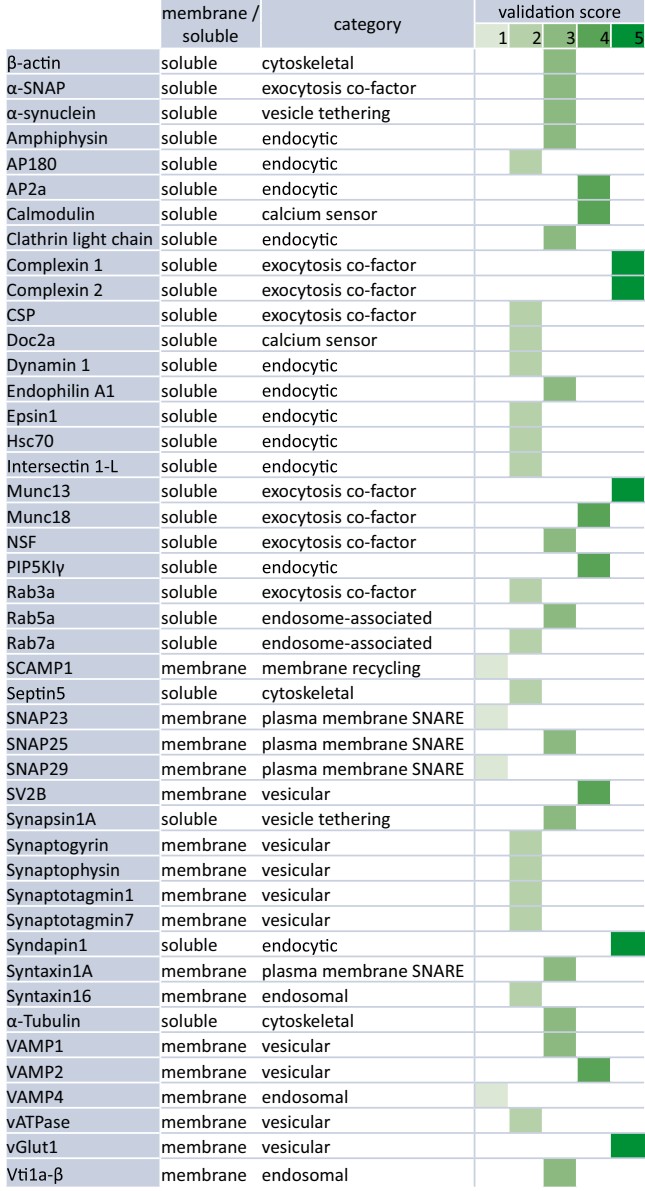

**Figure 1. Overview of proteins analyzed here and previous validation of the GFP chimeras we used, according to the literature.**

Protein categories according to their function and/or localization are indicated. We generated validation scores for all of the GFP-fused constructs we employed, as follows: 0) The tagged protein has not been tested before. Does not apply to any of the proteins we used. 1) The correct protein localization upon tagging is verified, but the function was not tested. 2) The correct protein localization upon tagging is verified, but function was difficult to test, due to the presence of the untagged protein. The appropriate function-related changes in the localization of the GFP-tagged proteins took place upon manipulations. 3) The appropriate protein function was verified for the tagged protein, typically in cell cultures (e.g., primary neuronal cultures). 4) The endogenous protein can be replaced by the tagged protein in cells in culture, with appropriate functional replacement. 5) The endogenous protein can be replaced by the tagged protein in living animals, with appropriate functional replacement. Most of the analyzed proteins have a score of 2 and more, meaning the correct localization and function of the tagged proteins have been shown previously. In detail, 4 proteins have a score of 1; 16 proteins have a score of 2; 14 proteins have a score of 3; 6 proteins have a score of 4; 5 proteins have a score of 5. The average score is 2.82. See Table EV1 for more details and for the references used.

We next aimed to determine whether the mobility of ectopically overexpressed mEGFP fusions would be different from that of the native proteins, or from that of knock-in proteins expressed at physiological levels (Appendix Fig S6). We compared our results with FRAP analyses of the following proteins. (i) Native synaptotagmin 1, tagged using a fluorescently conjugated antibody against its intravesicular domain, which we analyzed in the past (Kamin *et al*, 2010). (ii) Genomically labeled, knock-in vGlut1[Venus] (Herzog *et al*, 2011). (iii) Knock-in alpha-synuclein-GFP, expressed in mouse brains at levels comparable to those observed in human disease cases (Spinelli *et al*, 2014). In addition, we also compared the FRAP curves of the proteins that are known to be exceptionally enriched in synaptic vesicles, and are not present at substantial levels in any other synaptic compartment, to FRAP curves of synaptic vesicles, obtained after labeling the vesicles with an FM dye (Shtrahman *et al*, 2005). All of these measurements were similar or nearly identical to our observations (Appendix Fig S6), which allows us to conclude that in our experimental setup neither mEGFP fusion, nor overexpression influences protein distribution and mobility in a major fashion.

### The synaptic protein mobility correlates to their presence in synaptic vesicles, but not to their sizes

To extract biological insight from the FRAP experiments, we first considered the potential interactions of proteins with synaptic organelles, and especially with synaptic vesicles. A comparison of the mobility parameters of all proteins showed that proteins located in the synaptic vesicles and in endosomes have substantial immobile fractions in synapses (Fig 1E and F, Appendix Fig S4). Moreover, the FRAP time constants of the membrane proteins localized in synaptic vesicles correlated well with their enrichment in purified synaptic vesicles (Takamori *et al*, 2006, Appendix Fig S7). This confirmed the expectation that proteins that tend to localize to substantial levels in the plasma membrane had faster recovery kinetics than the proteins predominantly localized in the largely immobile vesicles (Appendix Fig S7). Interestingly, vesicular proteins also have higher time constants in axons, compared to other membrane proteins, although they are present in the axons mostly as proteins in the plasma membrane, and not as vesicles (Appendix Fig S8). An interesting case was that of VAMP4, whose recovery was substantially slower in axons than in the synapse, against the trend observed for most other membrane proteins. VAMP4 tends to be found in endosomes in the axon, but not in the synapse, as observed in our immunostainings for this protein (Appendix Fig S2), and therefore, its axonal FRAP values are probably influenced by the slow recovery of endosomes through active transport. Additionally, a strong correlation is observed between the time constant and the immobile fraction in synapses, but not in axons (Appendix Fig S9).

We then proceeded to test whether protein mobility can be linked to previously known protein characteristics such as structure, size, or localization. We found that for membrane proteins, both the time constants and the immobile fractions correlate positively with the number of transmembrane domains (Fig 4A, Appendix Fig S10A), in agreement with an expectation that the presence of multiple transmembrane domains would slow down diffusion (Kumar *et al*, 2010). For soluble proteins, however, we did not observe a correlation between molecular weight and the

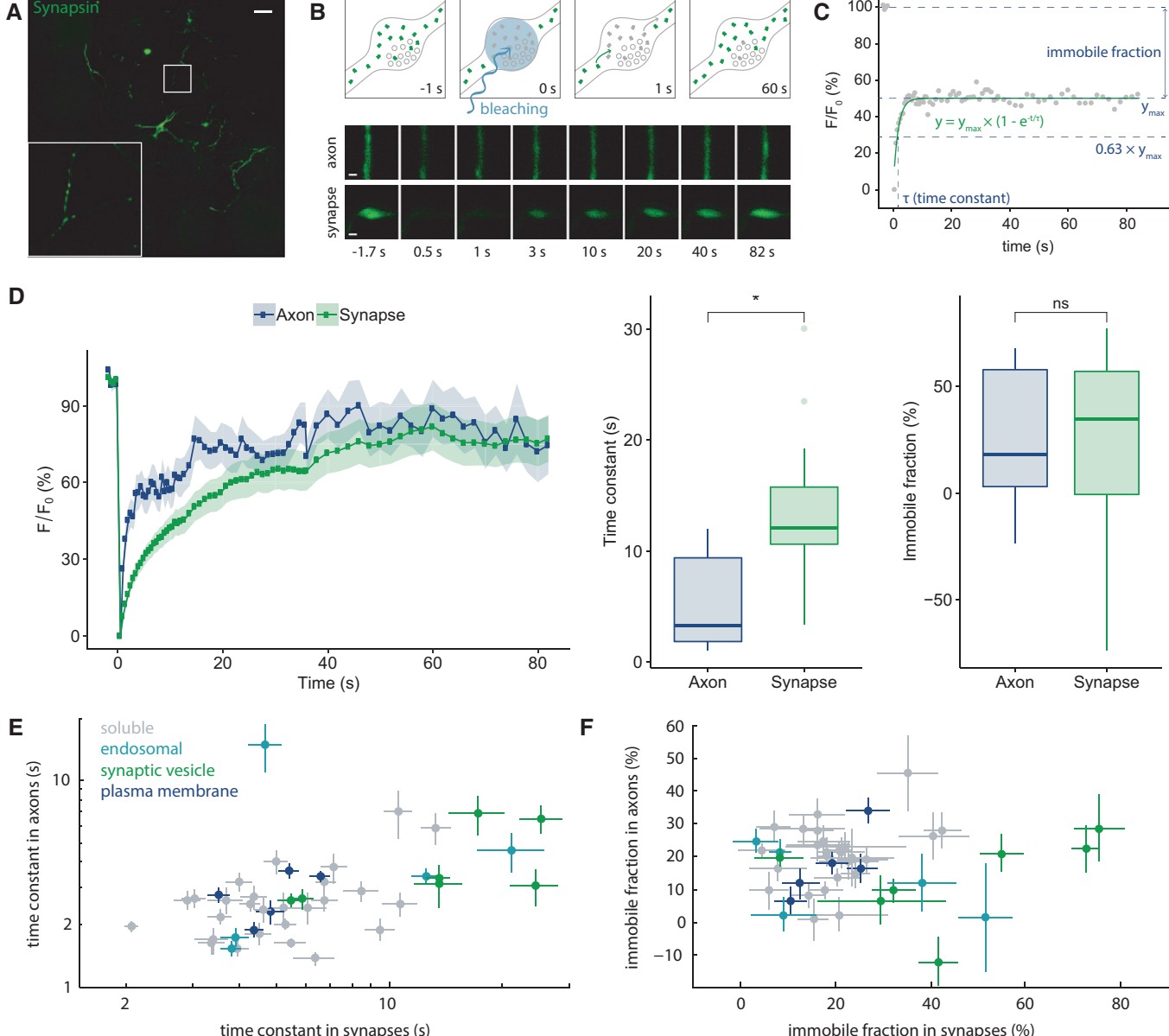

**Figure 2. An overview of FRAP experiments.**

A   Typical wide-field image of a neuron expressing the synaptic vesicle-binding protein synapsin coupled to mEGFP. Scale bar, 100 μm.

B   Top panels: a cartoon explaining the FRAP procedure. Fluorescent molecules are shown in green. The mEGFP molecules in a defined area are photobleached (gray molecules), and then, the entry of non-bleached molecules from the neighboring areas is measured. Middle and bottom panels: typical results in an axonal segment and in a synaptic bouton of a neuron expressing synapsin coupled to mEGFP. Scale bar, 500 nm.

C   An explanation of the FRAP analysis procedure. The FRAP recovery curves could be well fit by single exponential functions, which provide the time constant of recovery, as well as the fraction of the molecules that is not replaced (immobile fraction).

D   Exemplary results showing FRAP curves, time constants, and immobile fractions of synapsin in axons and synapses. Symbols indicate means ± SEM. The box plots are organized as follows: The middle line shows the median; the box edges indicate the 25th percentile; the error bars show the 75th percentile; and the symbols show the 90th percentile. Asterisk denotes significant difference. Wilcoxon rank-sum test with using the Benjamini–Hochberg procedure for multiple testing correction, FDR = 0.05. $N$ (axons) = 17, $N$ (synapses) = 24. Also presented in Appendix Fig S3.

E   Time constants of all analyzed proteins in axons and in synapses. The two parameters correlate significantly, albeit not very strongly ($R$ = 0.3182, $P$ = 0.04). This correlation is only observed for soluble proteins ($R$ = 0.6134, $P$ = 0.0005), and not for membrane proteins ($R$ = 0.0338, $P$ = 0.9086).

F   Immobile fractions in axons and synapses. No correlation was observed ($R$ = 0.0451, $P$ = 0.7769). Symbols indicate means ± SEM; all data are shown as box plots in Appendix Fig S3, numbers of replicates for each protein are shown in Appendix Fig S3, panels E and F are also presented in Appendix Fig S4 with protein names indicated next to symbols.

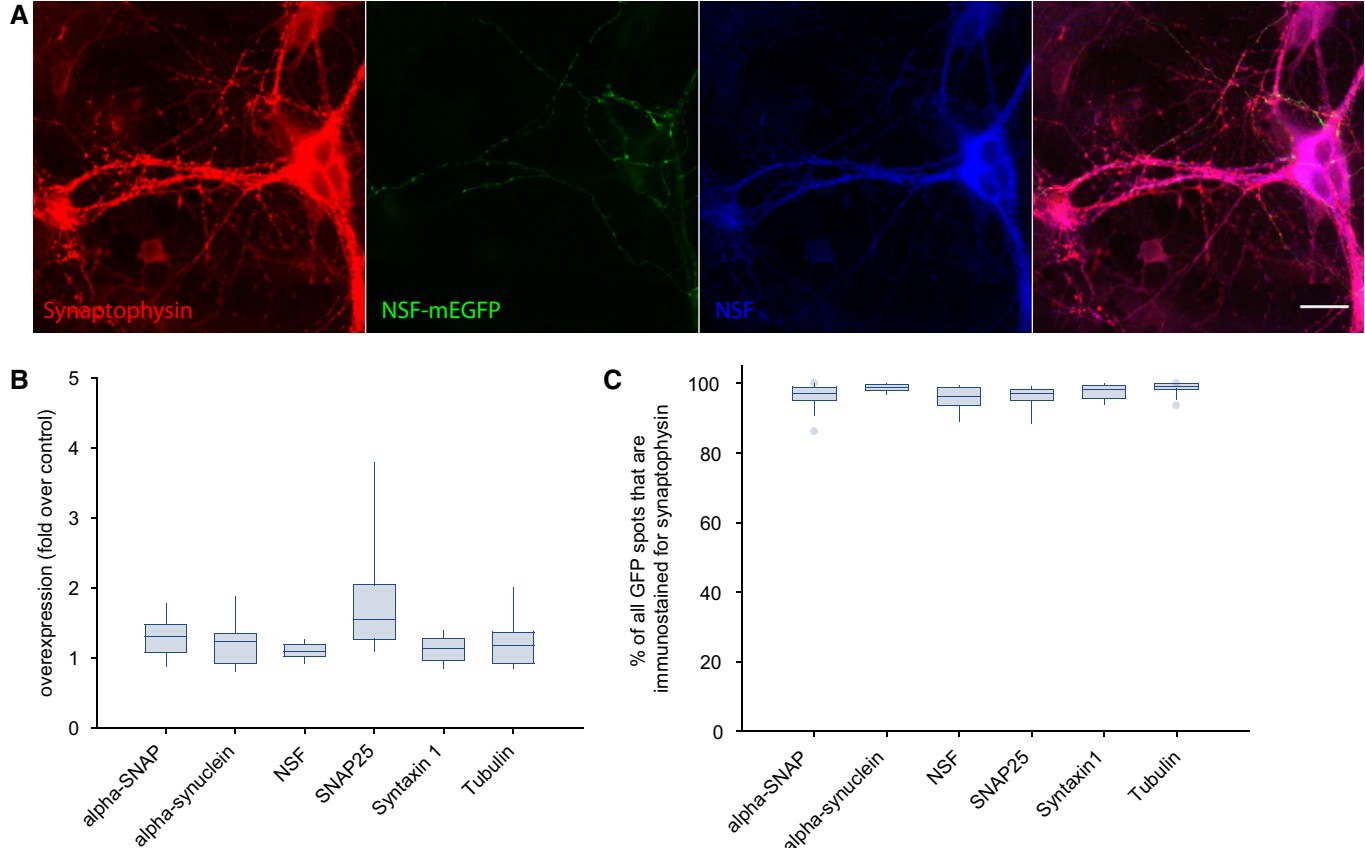

**Figure 3. Analysis of protein overexpression.**

A   Typical images of a neuron expressing alpha-SNAP fused to mEGFP (green), which was also immunostained for the same protein (blue), and for the synaptic vesicle marker synaptophysin (red), to detect synaptic boutons. Scale bar, 20 µm.

B   The levels of the proteins of interest were measured (relying on the immunostaining) in the transfected boutons, as well as in the non-transfected boutons (detected by the synaptophysin immunostaining). The overexpression levels are shown, obtained by dividing the immunostaining intensity in the overexpressing boutons by that in the non-overexpressing boutons. Only boutons with moderate expression levels have been considered in this work. $N = 3$ independent experiments, with ~6 independent fields of view (containing different neurons) per experiment.

C   Percentage of GFP-positive spots that are also immunostained for synaptophysin. $N = 3$ independent experiments, with ~6 independent fields of view (containing different neurons) per experiment.

Data information: The box plots were organized as follows: The middle line shows the median; the box edges indicate the 25th percentile; the error bars show the 75th percentile.

time constant (Fig 4B), as observed, for example, in bacteria (Kumar *et al*, 2010). Another simple observation was that membrane proteins, on average, were slower compared to soluble ones, which is in good agreement with the literature (e.g. Kumar *et al*, 2010). Both protein classes showed significantly higher time constants in synapses than in axons (Appendix Fig S10B), suggesting that the synaptic environment slows the movement of both protein classes.

**Synaptic protein mobility correlates to several other cell biology parameters, including structural features of the proteins and their lifetimes**

We next aimed to determine whether the amino acid composition or the presence of certain structural motifs can influence protein mobility. Such parameters have been linked to numerous features of the proteins in the past, such as their abundances or lifetimes (as

mentioned in the introduction), which makes such a comparison also interesting for the protein mobility.

We first compared the mobility parameters of the proteins to the amino acid composition of their sequences (Appendix Fig S11). Numerous correlations were found. For example, the synapse FRAP time constant was negatively correlated with the percentage of aspartate residues in the protein sequences (Appendix Fig S12A). As it bears a negatively charged side chain, aspartate is expected to increase protein solubility, which provides an explanation for this observation. Glutamate shows a similar trend, albeit this correlation was not statistically significant (Appendix Fig S11). In contrast, we observed strong positive correlations between the percentage of phenylalanine residues in the protein sequence and the synapse FRAP time constant (Appendix Fig S11). A similarly strong influence of the phenylalanine content was observed on the immobile fraction in synapses (Appendix Fig S11, Fig 4C). The effects of this amino

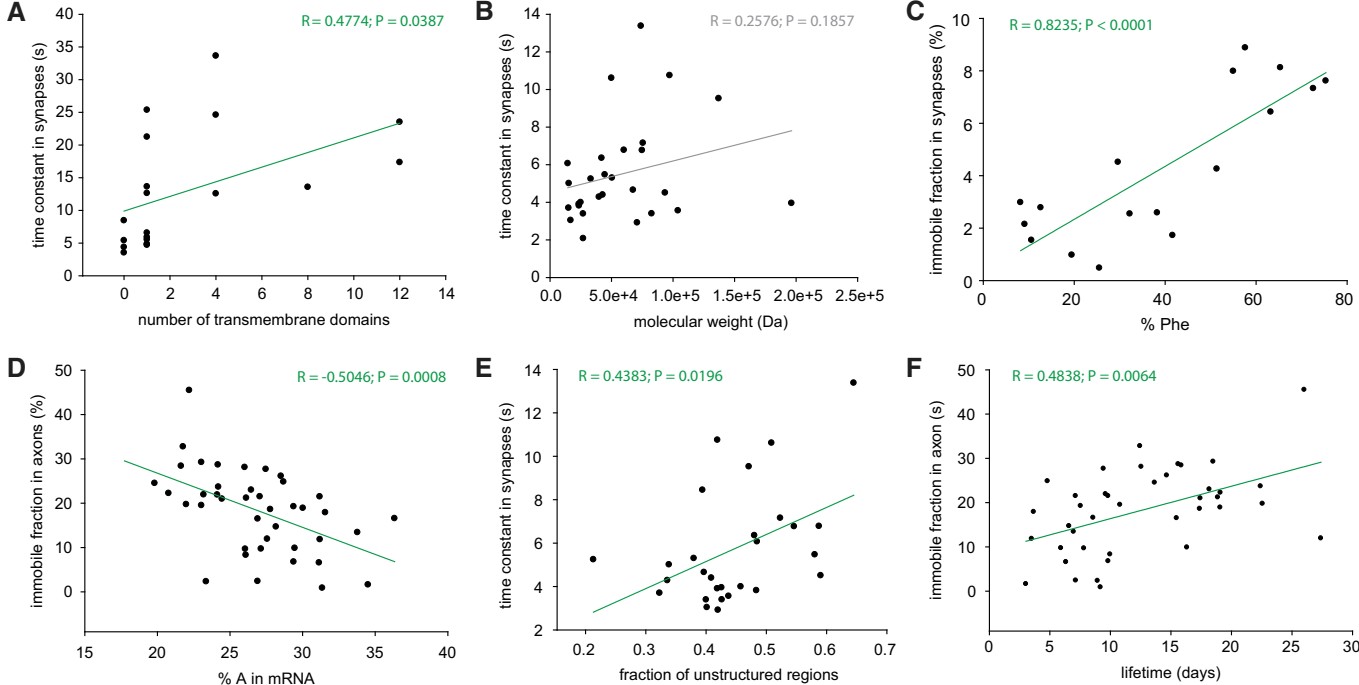

**Figure 4. Correlation of protein mobility to various parameters.**

A  Correlation of FRAP time constants in synapses with the number of transmembrane domains, for the different membrane proteins. A significant correlation can be observed, which agrees with the previous literature, and with the expectation that proteins with large numbers of membrane domains diffuse more slowly.

B  No correlation between the FRAP time constants in synapses and the molecular weight of the soluble proteins could be observed.

C  Correlation between the immobile fraction in synapses (for the membrane proteins) and the percentage of phenylalanine residues in the protein sequence.

D  Correlation between the immobile fraction in axons and the percentage of adenine in the mRNA sequences.

E  Correlation between time constants in synapses and the predicted fraction of unstructured coils in the protein structure.

F  Correlation between immobile fractions in axons and protein lifetimes. See Appendix Figs S10–S14 for more details.

acid were due to the contribution of membrane proteins, since no such correlation could be observed when only soluble proteins were considered (Appendix Fig S11). Moreover, the presence of other hydrophobic amino acids, as tryptophan, also correlated with low protein mobility in synapses. Overall, these observations are in agreement with the idea that proteins with higher numbers of transmembrane domains will contain proportionally more hydrophobic amino acids, while also being less mobile (Fig 4A, Appendix Fig S10A).

Having noted these correlations, we next turned to test whether such observations would also hold true at the mRNA level. We analyzed the correlation between mobility parameters and the percentage of different nucleotides in the respective mRNAs (Appendix Fig S13). We found significant negative correlations between the adenine percentage and the time constants in synapses and the immobile fractions, for both soluble and membrane proteins (Fig 4D, Appendix Figs S12B and S13). This is a relatively surprising observation, as none of the amino acids whose percentage correlates negatively with the protein mobility are coded by adenine-rich codons.

Overall, these observations suggest that structural parameters of the proteins (and possibly also of the respective mRNAs) may be linked to their mobility. To test this in a more direct fashion, we relied on predictions for the structured and unstructured regions in the proteins (from Mandad *et al*, 2018), and we correlated the mobility parameters with the fraction of the protein sequences that is structured (alpha-helix and beta-sheet) or unstructured (random coils). We

found that the more unstructured the protein is, the less mobile it is in both axons and synapses (Fig 4E, Appendix Fig S12C). This is also supported by a relatively high (albeit not significant) correlation between the percentages of proline residues and the time constants of soluble proteins in axons (Appendix Fig S11), since prolines tend to act as breakers of secondary structures (Chou & Fasman, 1974).

Finally, we also tested less expected connections, to different cell biology parameters, including the protein lifetimes (Fornasiero *et al*, 2018; Appendix Fig S14). Remarkably, we found a significant positive correlation between immobile fraction in axons and the protein lifetimes (Fig 4F). The biological relevance of this correlation seems relatively simple. Proteins with high immobile fractions in the axon would spend long time periods here, which implies that they are more slowly transported along the axon then other proteins. The time spent during this slow transport simply adds to the total lifetime of these proteins, which renders them longer-lived than rapidly transported proteins. However, the mechanisms behind this simple hypothesis still remain to be determined.

**Further considerations on alternative measurements**

One criticism that these experiments could face is the exclusive use of FRAP. Other technologies could, in principle, also have been used, including fluorescence correlation spectroscopy (FCS). However, FCS measurements are difficult to interpret in the complex 3D space of the synapse (Appendix Fig S15) and resulted

in values that did not conform to the existing literature (e.g., the apparent diffusion coefficient of mEGFP was $0.9 \pm 0.4 \ \mu m^2/s$, which is at least 10- to 20-fold below the expected value for this molecule in cell cytosol or in synaptic boutons; see Sadovsky *et al*, 2017 and references therein, and Spinelli *et al*, 2014). These difficulties are due to a number of factors that affect FCS interpretations. First, the exact intracellular viscosity at the measurement positions is unknown, as is also the local temperature during the acquisition time (due to laser-induced heating). Second, the 2D fitting model normally used for such measurements may not be the correct choice for modeling the data, as the axon, and most of the synapses, is substantially thinner than the excitation volume. This implies that diffusion in and out of the volume in the second dimension does not occur in the measurements. As a result, FCS measurements report diffusion coefficients that do not correspond to real protein behavior. To obtain true coefficients, complex modeling procedures that take synaptic and axonal geometry into account would be required.

An alternative option for identifying the molecule movement behavior is to employ single-molecule tracking. This technique results in high-precision data and has been performed for a few presynaptic molecules, including syntaxin 1 in hippocampal neurons (Ribrault *et al*, 2011) and in *Drosophila* synaptic boutons (Bademosi *et al*, 2017), or synaptotagmin 1 in hippocampal neurons (Westphal *et al*, 2008; Kamin *et al*, 2010). The procedures to perform single-molecule tracking are substantially more difficult than FRAP or FCS, as they have relied on complex labeling using quantum dots (Ribrault *et al*, 2011), on live STED imaging close to the performance limits of the respective instrumental setups (Westphal *et al*, 2008; Kamin *et al*, 2010), or on highly specialized analysis procedures, relying on photoconvertible fluorescent proteins (Bademosi *et al*, 2017). Therefore, such procedures have not been typically employed for many proteins in any given publication, and could not be employed efficiently for the 45 proteins analyzed here. Fortunately, a detailed analysis of our own work shows that the FRAP analysis can reproduce the results provided by single-molecule tracking in hippocampal neurons, as detailed below.

### Active organelle movement is a relatively rare event over the FRAP time course

Before proceeding with a thorough analysis of the FRAP data, one would need to consider the fact that FRAP does not differentiate between diffusive and active transport. Most proteins are delivered to synapses via both modes. For example, transmembrane proteins are transported actively as components of vesicles or endosomes, but they also diffuse passively in the plasma membrane. Therefore, the time constants observed in FRAP would report a mixture of the recovery of the molecule population found in the plasma membrane, and of the recovery of the population found in vesicles.

To estimate the extent to which active organelle transport would affect the FRAP observations we made, we aimed to estimate the fraction of recovery that can be caused by active transport of the organelle-bound proteins. We analyzed this experimentally by organelle-tracking experiments, again relying on the mEGFP chimeras presented above (Fig 5A). First, we calculated the fraction of each of the analyzed proteins found in organelles, as opposed to being distributed diffusely on the plasma membrane or in the cytosol. Second, we estimated the mobility of the organelles. This enabled

us to determine the fraction of each molecule that is present, at any one time, in mobile organelles, which would influence the timeline of the FRAP recovery. This analysis suggested that, as expected, < 2% of cytosolic mEGFP is present in moving organelles (presumably autophagosomes produced during neurite remodeling). In general, < 10–12% of the molecules were found in moving organelles (Fig 5B, left), even for vesicle proteins, mostly due to the fact that the organelles were immobile for most of the observation time. Immobility was defined as displacements at or below the levels observed in aldehyde-fixed samples. When mobility was observed, the average movement speed was up to ~1 μm/s (Fig 5B, middle), as expected from previous studies on neuronal organelle transport (Hirokawa *et al*, 2010). The duration of each movement episode varied from 1 to ~15 s (Fig 5B, right).

Overall, these experiments imply that the contribution of active organelle transport to the FRAP recordings we obtained is limited.

### A detailed analysis of the protein movement parameters obtained from FRAP imaging

The FRAP results showed here also do not report exact diffusion coefficients, but rather a comparable measure of apparent protein mobility. FRAP is notoriously difficult to analyze in terms of true molecular motion in synapses (Salvatico *et al*, 2015), and for the diffusion coefficients to be extracted from these data, the synapse geometry would also have to be considered. Albeit general FRAP interpretation models have been proposed (Kang *et al*, 2012; Blumenthal *et al*, 2015; Bläßle *et al*, 2018), they cannot be used with accuracy in a small and complex structure, as the synapse. When such models are used for our measurements, they underestimate the known motion behavior of GFP by at least 50- to 100-fold (diffusion coefficients of only ~0.2 $\mu m^2/s$ both in the axon and in the synapse, Appendix Fig S16). This is probably due to the fact that the general FRAP interpretation models are designed for situations in which molecules come from large cellular areas, from all directions, and are thus unable to account for the synaptic geometry.

To interpret the FRAP results one would need a realistic model, which considers the synapse organization. We sought therefore to simulate different particle motion behaviors, with different movement speeds, in realistic synaptic space in order to find the behaviors that most closely reproduced the FRAP results. An overview of the modeling procedure is presented in Fig 6. We started by generating a 3D synapse from electron microscopy data (Fig 6A). We then moved particles in this 3D synapse, at different speeds (Fig 6B). We hypothesized that the behavior of the proteins could be approximated as the diffusive movement of particles in the synaptic space (in the cytosol or in the plasma membrane). Afterwards, we transformed the particle motion in artificial FRAP movies (Fig 6C–E), which we then compared to the original FRAP data (Fig 6F), in order to find the models that best reproduced the biological measurements. We then used these models to determine the diffusion coefficients of analyzed proteins (Fig 6G) and to generate graphical representations of their movement (Fig 6H).

To obtain a 3D synapse model, we relied on serial-sectioning electron microscopy (*e.g.,* Schikorski & Stevens, 1997, 2001). We reconstructed 30 synapses and measured their different parameters, including surface, volume, active zone area, vesicle number,

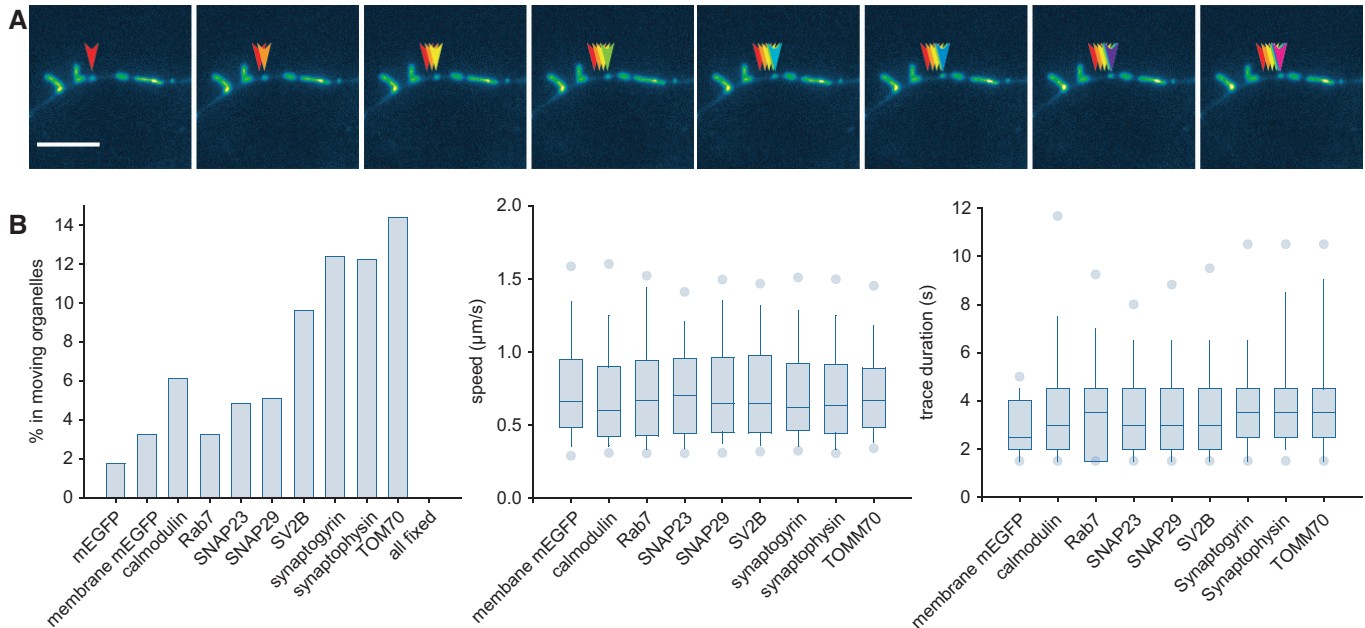

**Figure 5. A tracking analysis for synaptic organelles.**

A Neurons were transfected with a TOM70-mEGFP construct, to indicate mitochondria, and were visualized using an epifluorescence microscope. The arrowheads track the movement of one mitochondrion. Scale bar, 10 μm.

B The movement was analyzed using particle tracking, as indicated in Materials and Methods. The proportion of molecules found in moving organelles, the speed of movement, and the trace duration are indicated for the different proteins expressed. $N$ = 4–10 independent analyses per condition. The box plots were organized as follows: The middle line shows the median; the box edges indicate the 25th percentile; the error bars show the 75th percentile.

vacuole number and volume, and mitochondria volume (Appendix Fig S17). We then chose a synapse that was close to the overall average for most parameters, and constructed it *in silico* with the respective axon, to a total length of ~9 μm (Appendix Fig S17A). We then used this synapse to construct Monte Carlo models of particle movement, using the simplest possible assumptions.

For membrane proteins, we placed particles in the plasma membrane of the 3D synapse model and allowed them to move in random directions, with a given average single-step velocity. The particles were allowed to move with the same velocity in the axon and in the synapse, in accordance with previous super-resolution tracking experiments we performed on the synaptotagmin 1 (Westphal *et al*, 2008; Kamin *et al*, 2010). To be able to account for slow- or fast-moving proteins, we generated models with different single-step velocities, from 25 to 200 nm per movement step.

To generate artificial FRAP movies, we placed 1,000 particles in the 3D synapse model and allowed them all to move with the same single-step velocity. A measured point-spread-function (PSF) was convoluted with each of the particles, to thus mimic the movement of 1,000 GFP-tagged protein molecules. To obtain a FRAP situation, we bleached *in silico* areas similar to those bleached in the biological experiments. We then monitored the re-entry of fluorescent (non-bleached) particles in the bleached area. The artificial movies were then analyzed exactly as the real FRAP movies. The FRAP parameters that these *in silico* movies provided were similar to those observed for most membrane proteins in real experiments. For example, the slower FRAP recovery observed in the synapse, in comparison with the axon, was also observed in the models.

For the soluble proteins, we added one more level of complexity to the model, to account for protein binding to the vesicle cluster, which is a well-known phenomenon (Shupliakov, 2009; Denker *et al*, 2011a; Milovanovic & Camilli, 2017). Particles moved with the same single-step velocity in the axon and in the synapse, as above, but they were also allowed to interact with vesicles, and to be retained on their surfaces (Fig 6B). We then combined different single-step velocities (25–250 nm per movement step) with different vesicle-retention times (from 1, meaning no retention, to 200, with particles staying on vesicles for ~200 movement steps, before coming off and moving again). This accounts for many different behaviors, such as slow or fast movement, as well as weak or strong interactions to the synaptic vesicles. The *in silico* FRAP data we obtained overlapped well with the FRAP results obtained in living neurons. Every measured protein behavior was reproduced by a specific combination of velocity and vesicle retention time, with the average difference between the measured and modeled FRAP time constants being ~5% (for FRAP in the synapse) and ~9% (for FRAP in the axon). For each combination, we calculated a diffusion coefficient using the Einstein–Smoluchowski equation and assigned the coefficients to proteins whose behaviors (FRAP recoveries) were best reproduced by the corresponding models.

## A series of validations for the FRAP interpretations

The models presented above are minimalistic in nature. For example, single membrane molecules can exhibit alternating slow and rapid movement phases (Freeman *et al*, 2018), but only the average movement speed is reported here. The models also replace complex

behaviors of soluble molecules, which may include repeated binding and unbinding to several different proteins within the cluster, with one single parameter: binding to the vesicle cluster. Some proteins, such as actin, may not even bind vesicles directly, but rather actin strands or synapsin within the cluster. In spite of these shortcomings, the models should be able to address the average behavior of

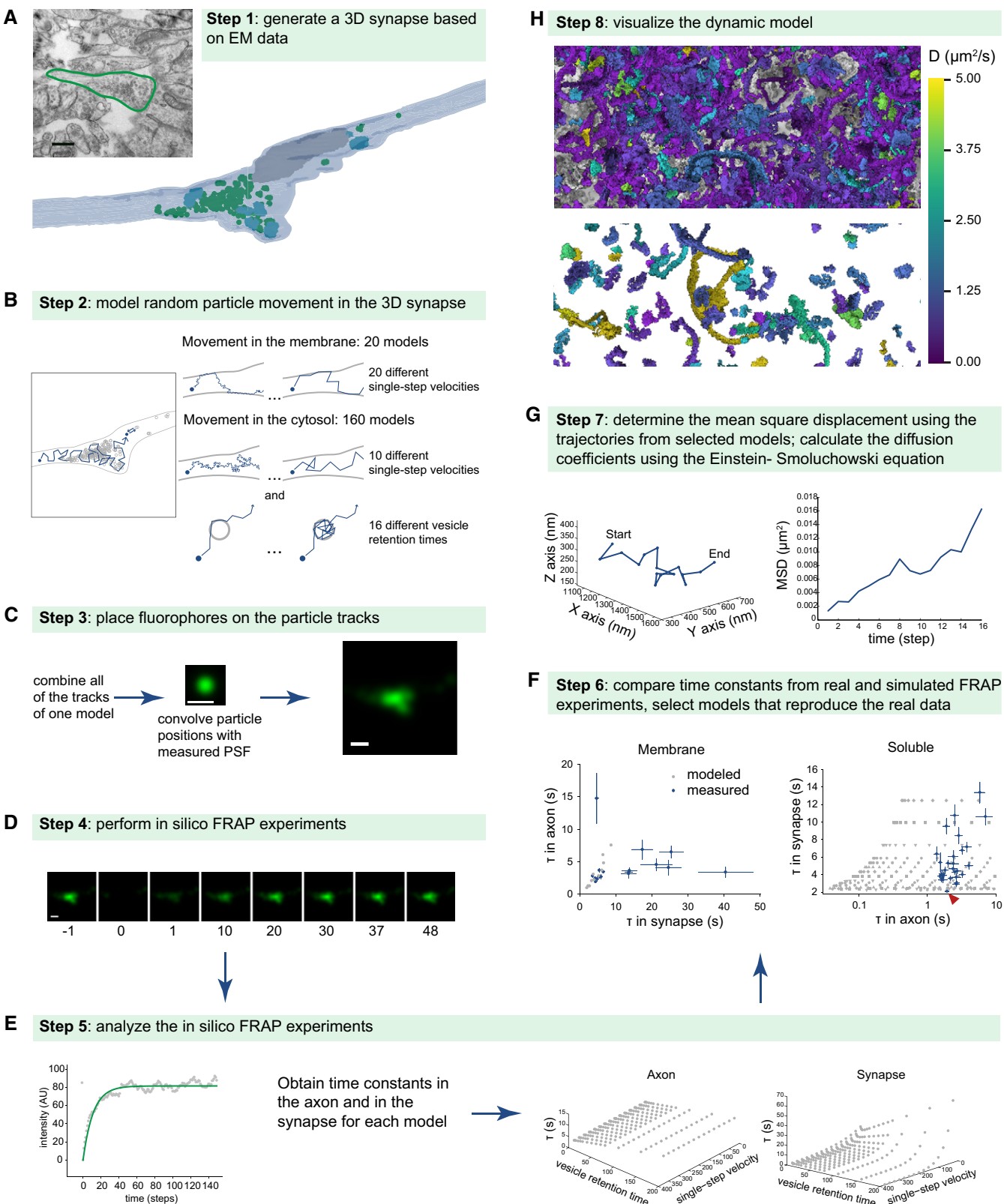

**Figure 6.**

**Figure 6. The procedure of creating the dynamic model.**

A   Protein movement was modeled in a realistic synaptic 3D space, obtained from electron microscopy. Also shown in Appendix Fig S17.
B   We allowed particles to move randomly with set average displacement steps (for both movement in the plasma membrane and in the cytosol) and vesicle retention times (only in the cytosol). Each model had a unique combination of the velocity and the vesicle retention time.
C   All generated tracks from one model were then combined, and particle positions were convoluted with a measured PSF to produce simulated fluorescent images.
D   By eliminating the fluorescence in a region similar in size to the bleaching region of real FRAP experiments, *in silico* FRAP movies were generated.
E   These were then analyzed in the same manner as the real FRAP experiments.
F   We selected the models reproducing the behavior of the proteins measured in live neurons by comparing the time constants obtained in the simulated FRAP experiments with the real ones. An interesting point can be observed in the right panel: the lowest blue spot (red arrowhead) represents EGFP. This is the only protein whose behavior overlaps with the lowest series of model behaviors (gray spots), which represents free motion in the synapse, without vesicle binding.
G   To obtain diffusion information we calculated for these models the mean square displacements (MSD) from the tracks, and we then derived from them diffusion coefficients (D) using the Einstein–Smoluchowski equation.
H   The diffusion coefficients are presented in Table EV2, while the model tracks are used to generate animated model representations (see EV movies). This panel indicates protein motion (diffusion coefficients) inside (top) and outside (bottom) the vesicle cluster in a color scheme. The top panel shows mostly vesicle-bound proteins, which have low mobility (in purple), while the space outside of the vesicle cluster contains mostly mobile proteins (green-yellow colors).

the different protein species. To test whether these models indeed report the true average behavior of the proteins, we measured or noted several independent parameters.

First, we found that the models correctly predicted that free mEGFP does not interact with synaptic vesicles (Fig 6F).

Second, the models also predicted the distribution of each protein in the synapse versus the axon. This correlated well with values obtained by immunostaining neuronal cultures for the different proteins, with a coefficient of determination ($R^2$) of ~0.7 (Fig 7A–C).

Third, we used the models to determine the copy number of proteins per synaptic vesicle, for the proteins that are known to reside in vesicles. This reproduced well the copy numbers measured in the past by protein biochemistry (Takamori *et al*, 2006) ($R^2 > 0.9$; Appendix Fig S18).

Fourth, we used the protein positions provided by the models to reconstruct stimulated emission depletion (STED) images for the different proteins, and we compared them with real STED images, obtained from immunostaining experiments (Fig 7D and E). The model images reproduced well the spot size and spot intensity distributions from the real images (Fig 7D–I; see Appendix Fig S2 for details on all STED immunostainings).

Fifth, the models were able to predict the fraction of membrane proteins found in the synapse, which correlated well with the enrichment of these proteins in the synaptic vesicles (Takamori *et al*, 2006; Appendix Fig S19).

Sixth, we similarly could predict the enrichment of the soluble proteins in the synaptic vesicle cluster, which correlated well with the enrichment of soluble proteins on purified synaptic vesicles (Takamori *et al*, 2006; Appendix Fig S20).

### Different diffusion coefficients for the proteins analyzed here

The validations indicated above suggest that, although we used bulk measurements (FRAP) and very simple interpretation models, our results were sufficiently robust to reproduce, among other parameters, previous results on protein distributions at the nanoscale. We therefore determined diffusion coefficients for the different proteins, from the models indicated above (Table EV2). To account for possible errors in the modeling, for each protein we averaged the diffusion coefficients of the three to five models whose FRAP values were closest to those of the respective protein. The resulting means and error bars are shown in Fig 8.

They provided two further lines of validation:

First, the diffusion of several proteins conformed to previously measured values (Fig 8A). mEGFP, with a modeled diffusion coefficient of ~20 $\mu m^2/s$, is well within the range measured in many other systems (15–26 $\mu m^2/s$, with an average of ~21 $\mu m^2/s$; see Sadovsky *et al*, 2017 and references therein). Synaptotagmin 1 had a modeled diffusion coefficient of ~0.11 $\mu m^2/s$, very close to the one measured by live super-resolution tracking of antibody-tagged molecules, ~0.095 $\mu m^2/s$ (Kamin *et al*, 2010). Similarly, the axonal diffusion coefficient of syntaxin 1 (~0.22 $\mu m^2/s$) was close to the value observed by tracking quantum dot-tagged molecules (Ribrault *et al*, 2011) in axons (0.2 $\mu m^2/s$). The synaptic diffusion coefficient of syntaxin was also similar to the average coefficient measured in this work (Ribrault *et al*, 2011), albeit it was ~20% larger. This may be due to the fact that some quantum dot-tagged molecules may have been slowed in the synapse by problems with quantum dot penetration in the synaptic cleft (Lee *et al*, 2017).

Second, from the behavior of synaptic vesicle proteins we could calculate the diffusion coefficient of whole synaptic vesicles (see Materials and Methods). The resulting value, of ~0.01 $\mu m^2/s$ (Fig 8B), is well within the range measured for synaptic vesicles in hippocampal neurons in the past (average of ~0.0138 $\mu m^2/s$, taking into account the studies in Jordan *et al*, 2005; Shtrahman *et al*, 2005; Yeung *et al*, 2007; Westphal *et al*, 2008; Kamin *et al*, 2010; Lee *et al*, 2012; Rothman *et al*, 2016).

As expected from our analysis of basic FRAP parameters (Fig 4, Appendix Fig S10), the diffusion coefficients confirmed that soluble protein movement was only loosely influenced by protein size and that membrane protein movement was significantly affected by the number of transmembrane domains (Appendix Fig S21).

More interestingly, the modeling analysis we performed enabled us to produce the first realistic movies of nanoscale protein movement in the synapse. We modeled the molecular motion of all of the analyzed proteins, in a realistic synaptic setting. Movie EV1 shows several thousand cytosolic protein molecules moving within a small region next to the vesicle cluster, while Movie EV2 shows membrane proteins moving in the plasma membrane above the same region (proteins are represented according to the legend in Appendix Fig S22). Many other graphic representations could be made, including views of molecule mixing in the synapse, for the soluble (Movie EV3) or membrane proteins (Movie EV4).

# Discussion

We provide here a first comparative study of protein mobility in the synaptic bouton, encompassing 45 different proteins, from different types and classes. Our results confirm several expectations, including the lower mobility of membrane proteins when compared to soluble proteins, or the lower mobility of virtually all proteins in the synapse, when compared to the axon. Other expected observations

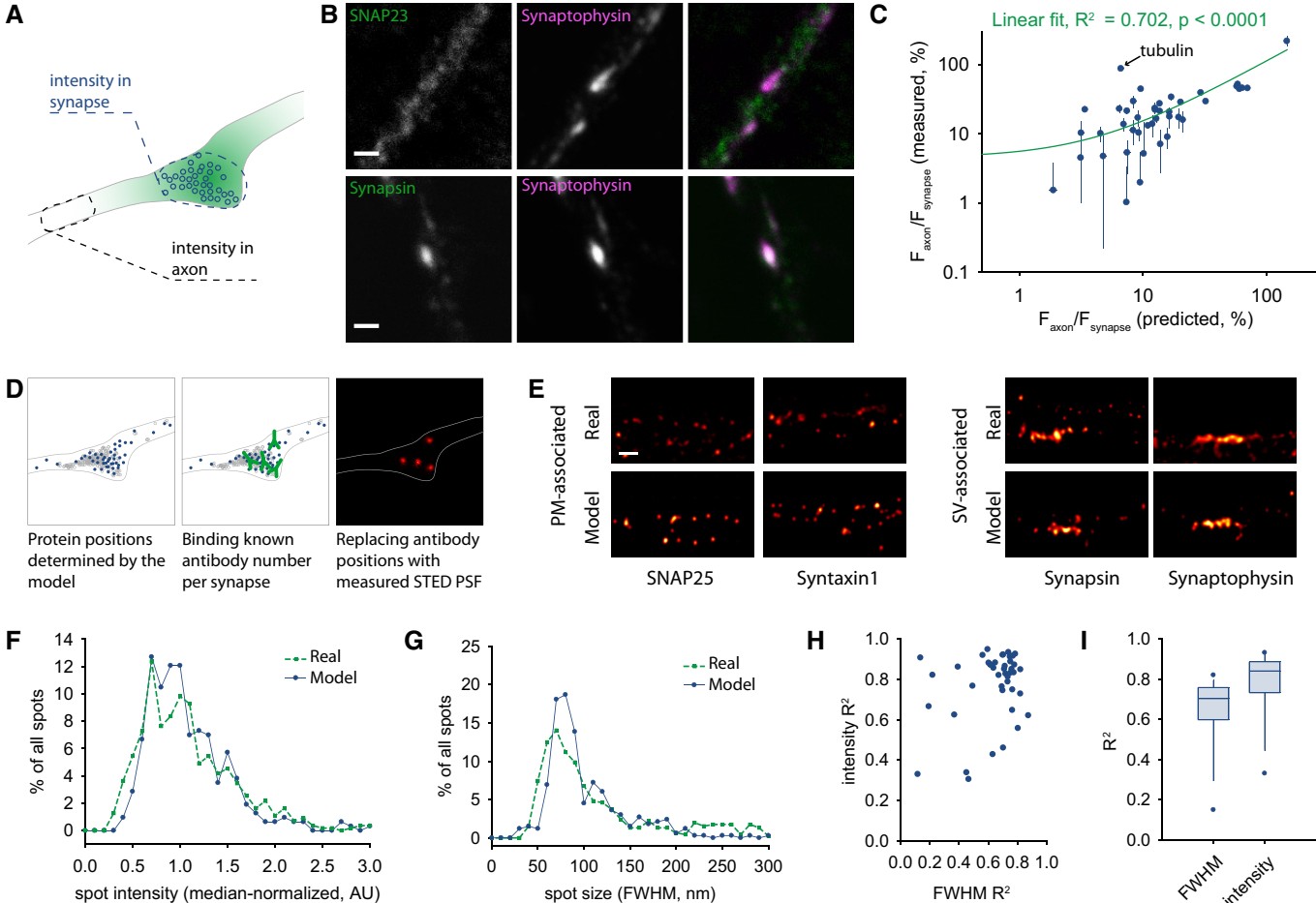

**Figure 7. Modeling validations.**

A   We analyzed the protein distribution in synapses versus the neighboring axonal segments, both *in vivo*, by immunostaining and confocal microscopy, and in the *in silico* models, by interpreting the particle positions.

B   Examples of immunostained proteins, which are present mostly in synapses (synapsin) or are distributed both in synapses and in axonal areas (SNAP23). Synapses were identified by co-immunostaining for the synaptic vesicle marker synaptophysin and for the active zone marker bassoon (not shown here). Scale bar, 500 nm.

C   A correlation of the fluorescence ratios between the axonal segments and the synapses, predicted by the model and in the biologically measured data. The correlation is highly significant, with a coefficient of determination ($R^2$) of ~0.7. This increases to 0.8 when tubulin is removed; for tubulin, we can only model the diffusion of the free molecules, but the immunostaining reveals both the free molecules and the microtubules, implying that a poor correlation is expected for this protein. Our analysis predicts that larger amounts of free tubulin are found in the synapse, versus the axon. This is very likely, for example, due to the larger volume of the synapse, so that the model prediction appears reasonable, although it cannot be tested in this experiment. Symbols indicate mean ± SEM from at least 20 neurons, from at least 2 independent experiments.

D   We relied on the model-suggested protein distributions to generate putative super-resolution images for the different proteins. Defined numbers of protein positions, corresponding to the number of antibodies that can be accommodated in these synapses for each protein (from Richter *et al*, 2018) were convoluted with a measured stimulated emission depletion (STED) PSF, thereby resulting in STED images for these proteins.

E   Typical model STED images are compared to real ones, obtained by STED imaging of immunostainings. Scale bar, 500 nm. Here, we only show real (immunostained) synapses that correspond in overall size to our model synapse.

F, G   The spot sizes (as full width at half maximum, FWHM) and intensities were analyzed in model synapses (using 100 different random STED images of synapses for each protein of interest) and in the real synapses (using all synapses, irrespective of size, in at least 15 different neurons from at least two independent experiments).

H, I   We analyzed the correlation between the spot size and intensity distributions, in the models and in the real synapses. These are shown as a scatter plot in h, or as box plots in i (the middle line shows the median; the box edges indicate the 25th percentile; the error bars show the 75th percentile; the symbols show the 90th percentile). Overall, the models correlate very well with the real data, with coefficients of determination ($R^2$) of ~0.7–0.8. The number of elements quantified here is identical to the number of proteins analyzed in the STED microscopy experiments (44).

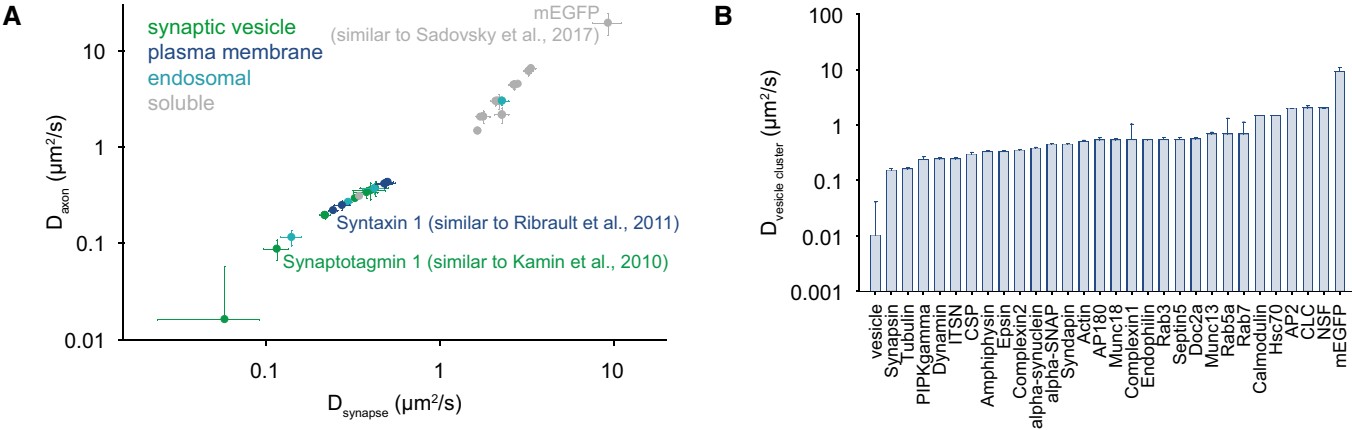

**Figure 8. Modeling results.**

A The diffusion coefficients obtained for the different proteins, in the synapse or in the axon. The symbols show the means ± SEM from the *in silico* models that best reproduced the data, corresponding to the range of values of 3–5 models. The values for all proteins are shown in Table EV2.

B Same as a, but for the diffusion in the vesicle cluster. For all *bona fide* synaptic vesicle proteins, this is represented by the diffusion of the vesicles themselves (back-calculated from the FRAP results of the vesicle proteins as explained in Materials and Methods). Bars show mean ± SEM, from the *in silico* models that best reproduced the data, corresponding to the range of values of 3–5 models.

were that the movement rates of the same proteins in the axon and in the synapse correlated little, presumably due to the different conditions encountered there, or that the mobility of soluble proteins was only little controlled by their molecular size. Several other observations could be made, including relations between protein mobility and structural parameters, mRNA composition, or protein lifetimes.

Our measurements, as indicated above, were performed with the caveat that the proteins we measured were more abundant than under normal (wild-type) conditions. The overexpression levels we observed were mild, but they may nevertheless contribute to artifacts, namely to an over-estimation of the mobility of individual proteins. If one protein is expressed too highly, its copy numbers saturate all binding to interacting partners, and the un-bound molecules end up moving randomly in the synapse, presumably at the highest possible speeds. The various validations we performed suggest that this is not a major problem. At the same time, our reported values should be taken as maximal mobility estimates, due to this issue. Native (non-tagged) proteins have not been investigated often, with only a handful of studies available. Several such studies were reproduced well by our data (Appendix Fig S6). At the same time, a FRAP measurement of knock-in Munc13 provided a substantially lower mobility (Kalla *et al*, 2006), with the shortest time constant measured in cultured mouse cortical neurons being around ~3 min, as opposed to a few seconds in our measurements. This difference probably has both technical and biological grounds. The previous work imaged the synapses at intervals of a few minutes, to avoid photobleaching. Analyzing our Munc13 data at 20–30 s of intervals (as opposed to two times per second, as in our original data) raised the time constant from ~4 to ~30 s. Analyzing such data every few minutes would presumably result in an even longer time constant. Also, the previous work bleached multiple boutons in the same area, which probably resulted in bleaching a considerable proportion of the fast-moving molecules in the respective axons, which will reduce the fluorescence recovery.

Nevertheless, it is still possible that Munc13 mobility is particularly sensitive to overexpression.

In spite of these caveats, several conclusions could nevertheless be drawn. First, synaptic protein mobility seems to be influenced by the interaction of the proteins with the vesicles. For soluble proteins, it has been hypothesized that strong interactions to the vesicle cluster cause their enrichment in synapses (Shupliakov, 2009; Denker *et al*, 2011a; Milovanovic & Camilli, 2017). This was observed especially for synapsin (Benfenati *et al*, 1989; Takamori *et al*, 2006; Milovanovic *et al*, 2018), whose slow movement in synapses was paralleled by strong binding to vesicles. This effect was even more strongly visible for membrane proteins (Appendix Fig S7) and is mostly explained by the fact that molecules that are more highly enriched in synaptic vesicles are present at lower levels on the plasma membrane. This implies that large fractions of these proteins will recover slowly during FRAP, through the infrequent active transport of synaptic vesicles (Fig 5). This will result in large time constants for the respective proteins. However, this is not the only explanation for this observation. The time constants of *bona fide* synaptic vesicle proteins are also higher in the axon, when compared to non-vesicular proteins (Appendix Fig S8). As all of these molecules are found in axons mainly as molecules fused to the plasma membrane, an explanation based on the transport of synaptic vesicles seems unlikely. A potential solution to this question is that synaptic vesicle proteins may diffuse in the axon in the form of assemblies composed of multiple molecules. This issue has been discussed for several decades (see e.g., Ceccarelli & Hurlbut, 1980; Haucke *et al*, 2011; Rizzoli, 2014), and it is still open for further interpretation. However, a series of recent observations, made mainly through super-resolution imaging of fused synaptic vesicles, suggested that such assemblies are indeed present in the axon, and may even be the dominant form in which vesicle proteins are found in the axonal compartment (Richter *et al*, 2018; Truckenbrodt *et al*, 2018; Seitz & Rizzoli, 2019).

Second, soluble unstructured proteins also appeared to move more slowly in synapses. This observation is especially interesting in the context of a recently proposed mechanism of synaptic vesicle cluster segregation. It has been suggested that synaptic vesicles, together with synaptic vesicle binding proteins, form a distinct liquid phase via liquid–liquid phase separation within the synapses (Milovanovic & Camilli, 2017; Milovanovic *et al*, 2018). By definition, material exchange between liquid phases is slower than free diffusion; therefore, it is expected that soluble proteins of synaptic vesicle cluster would have slower recovery rates. Since the presence of multiple disordered coils is one of the main structural characteristics of proteins known to take part in liquid phase separation, our observations fit very well with this model.

Third, several correlations could be found to the presence of different amino acids in the protein sequence, or to the presence of particular nucleotides in the mRNA sequence. While the correlations to specific amino acids were relatively easy to interpret, as mentioned in Results, the links to mRNA composition are less obvious. Different parameters of protein homeostasis have been linked to the mRNA composition in mammalian cells, and especially to the mRNA secondary structure (Kudla *et al*, 2009) or to the GC contents (Kudla *et al*, 2006). At the same time, the mRNA composition has been suggested to control the folding conformation of specific proteins (Zhou *et al*, 2013; Fu *et al*, 2016). It is still unclear whether the relations between mRNA composition and cell biology parameters are causative in nature (Arhondakis *et al*, 2008), but they are sufficiently strong to enable reasonable predictions of protein abundance, lifetime, and translation rate (Mandad *et al*, 2018). Overall, it is therefore not entirely surprising that protein mobility also correlates with mRNA composition, albeit it is difficult to explain why a high percentage of adenine correlates with higher mobility. One hypothesis could be based on the observation that proteins related to specialized function, including synapse formation, are encoded by GC-rich genes (Gingold *et al*, 2014; Fornasiero & Rizzoli, 2019). In contrast, proteins involved in cell proliferation and in general cellular metabolism are encoded by AU-rich genes. This implies that "less synaptic" proteins would have mRNAs containing higher adenine percentages than *bona fide* synaptic proteins. The former would interact less with synaptic vesicles or other synaptic components, and would therefore be more mobile than true synaptic proteins. This hypothesis is supported by the fact that the top adenine-containing targets, which influence mostly these correlations, are SNAP23, Rab5, VAMP4, and SNAP29, trafficking molecules that are not specific to synapses in any fashion. At the opposite end, the top adenine-lacking proteins are the synaptic vesicle markers synaptophysin, synaptogyrin, and vGlut (glutamate transporter), along with the endocytosis cofactor epsin, further confirming this hypothesis.

Finally, we analyzed thoroughly the FRAP data, to provide diffusion coefficients for the different proteins. These coefficients were validated by several types of measurements, as described above, and should provide a good starting point for models of synaptic physiology. We are confident that laboratories specialized in neuronal and synaptic modeling could exploit our entire dataset by introducing the different protein amounts and mobilities in multi-reaction synaptic models (as, e.g., in Gallimore *et al*, 2018). Importantly, our data could be compared and combined with any dataset on hippocampal cultured synapses, which are a commonly used experimental model, for which large numbers of functional datasets are available.

We conclude that our work provides a novel resource for the analysis of synaptic function, which should enable synaptic modeling with substantially higher precision than in the past.

# Materials and Methods

### Cell culture and transfections

Primary hippocampal neuronal cultures were obtained from newborn (P1-P3) rats as previously described (Banker & Cowan, 1977; Kaech & Banker, 2006). Cells were grown in 12-well plates on glass coverslips in Neurobasal-A medium (Gibco, Paisley, Scotland), pH 7.5, at 37°C in 5% $CO_2$.

Cells were transfected using either calcium phosphate or lipofectamine transfection. Transfections with calcium phosphate were performed after 7–8 days *in vitro*, using the ProFection® Mammalian Transfection System Calcium Phosphate Kit following a protocol according to the manufacturer's recommendations with slight modifications described previously (Truckenbrodt *et al*, 2018).

Lipofection was performed after 2–6 days *in vitro*. Coverslips with cells were placed into 400 µl of DMEM (BioWhittaker, Lonza, Verviers, Belgium) complemented with 10 mM $MgCl_2$ and incubated for 30 min at 37°C in 5% $CO_2$. 50 µl of a transfection mix containing 2 µl of Lipofectamine® 2000 reagent (Invitrogen, Carlsbad, CA, USA) and 1 µg of plasmid DNA in Opti-MEM® (Gibco, Paisley, Scotland) medium were added per coverslip. After 20-min incubation at 37°C in 5% $CO_2$, cells were washed three times with DMEM (BioWhittaker, Lonza, Verviers, Belgium) complemented with 10 mM $MgCl_2$ and placed into original wells with Neurobasal-A medium (Gibco, Paisley, Scotland).

### DNA plasmids

For ectopic expression of fluorescently labeled proteins, plasmids coding for proteins of interest fused to a monomeric variant of enhanced GFP (A206K mutant, mEGFP) or EGFP were used. See Table 1 for mRNA reference sequences, backbones used, and sequences of linkers between the fluorescent tag and the proteins of interest of the plasmids used in the FRAP experiments. Information on the chimeric proteins is also shown in Appendix Fig S3.

Synthesis and cloning of sequences coding for mEGFP-beta-Actin, mEGFP-alpha-SNAP, Amphiphysin-mEGFP, mEGFP-AP180, mEGFP-AP2 alpha-C, Calmodulin1-mEGFP, Clathrin light chain b-mEGFP, Complexin 1-mEGFP, Complexin 2-mEGFP, mEGFP-CSP, Dynamin-mEGFP, EndophilinA1-mEGFP, Epsin-mEGFP, mEGFP-Hsc70, mEGFP-Intersectin 1-L, Munc13-mEGFP, NSF-mEGFP, Rab3a-mEGFP, Rab7a-mEGFP, mEGFP-SCAMP1, mEGFP-Septin5, SNAP23-mEGFP, SNAP29-mEGFP, SV2B-mEGFP, Synaptogyrin-mEGFP, Synaptotagmin 1-mEGFP, Synaptotagmin7-mEGFP, Syndapin1-mEGFP, mEGFP-Syntaxin16, VAMP1-mEGFP, VAMP4-mEGFP, vDlut1-mEGFP, and vATPaseV0A1-mEGFP into a mammalian expression vector pcDNA3.1 were ordered at GenScript® (Piscataway, NJ, USA). N- or C-terminal position of fluorescent tag was chosen based on information available on influence of, respectively, positioned tag on protein localization or function favoring that with minimal reported effect. Plasmids coding

for Munc18-EGFP, Syntaxin 1-mEGFP, and VAMP2-EGFP were previously described (Vreja *et al*, 2015). Plasmids coding for Doc2a-EGFP, Rab5-EGFP, mEGFP-SNAP25, and Synaptophysin-EGFP were produced by eliminating stop codons from previously described plasmids (Vreja *et al*, 2015). For this, primers listed in Table 2 were used. Primers were designed to anneal to coding sequences and contain overlapping regions to be used in the Gibson Assembly reaction. In case of Doc2a, Rab5, and Synaptophysin, only one pair of primers was used to amplify the region of interest (entire plasmid except for stop codon containing linker) with overlapping regions coding for new linkers. For SNAP25, three pairs of primers were synthesized to amplify the vector, mEGFP, and SNAP25-coding sequences. Plasmid coding for Vti1a-β-mEGFP was made by cloning the Vti1a-β-coding sequence from a previously described plasmid (Vreja *et al*, 2015) into an mEGFP-containing vector purchased from GenScript (Piscataway, NJ, USA).

PCR amplification was done using Phusion polymerase (New England Biolabs, Ipswich, MA, USA) according to recommendations of the manufacturer. Amplified fragments were used for assembly of the plasmids in the Gibson Assembly (New England Biolabs, Ipswich, MA, USA) reaction, performed according to recommendations of the manufacturer.

For expression of a control protein mEGFP, an empty pmEGFP-N1 vector was used, which was a gift from Prof. Dr. Reinhard Jahn (Max Planck Institute for Biophysical Chemistry, Göttingen, Germany). As a control for membrane proteins, a plasmid coding for mEGFP fused to palmitoylation sites of SNAP25 (amino acids 1–14, 80–142, and 203–206) was used. Plasmids coding for EGFP-Tubulin and EGFP-PIP5KIgamma were obtained through Addgene (plasmid numbers 30487 and 22299, respectively); EGFP-Synapsin and mCherry-Synapsin were gifts from Prof. Dr. Flavia Valtorta (San Raffaele Vita-Salute University, Milan, Italy) and have been previously described (Pennuto *et al*, 2002; Verstegen *et al*, 2014); alpha-Synuclein-EGFP was a gift from Prof. Dr. Tiago F. Outeiro (University Medical Center Göttingen, Germany).

### FRAP experiments

A TCS SP5 confocal microscope (Leica, Wetzlar, Germany) equipped with an HCX Plan Apochromat 100× 1.40 oil immersion objective was used for the imaging. The 488 nm line of an Argon laser was used for imaging of EGFP. Neurons were used for the FRAP experiments 3–7 days after calcium phosphate transfection or 6–14 days after lipofection (at least 11–14 days in culture). The culture medium was replaced by pre-warmed Tyrode's solution (124 mM NaCl, 2.7 mM KCl, 10 mM Na$_2$HPO$_4$, 2 mM KH$_2$PO$_4$, pH 7.3). The temperature of the imaging chamber system was maintained at 37°C. Synapses were located manually based on their morphology, preferably distal synapses, and axonal segments were used in FRAP experiments. For imaging of single synapses, 48× zoom and a 128 × 128 pixel resolution were used; same settings were used for imaging of axonal segments. Before bleaching, 4 control images were taken, and then, the region of interest was bleached for 80 ms with laser intensity of 50 μW at 488 nm, 14 μW at 496 nm, and 15 μW at 476 nm. After bleaching, 24 images were taken every 0.5 s, then 24 images every 1 s, and 24 images every 2 s. Additionally, images with the same time settings, but using 0% laser intensity for bleaching, were acquired to be used for acquisition bleaching correction.

### FRAP image analysis

The FRAP movies were analyzed automatically using custom-written MATLAB (The MathWorks Inc, Natick, MA, USA) routines. After loading all frames, the FRAP region was automatically determined, by comparing the last pre-FRAP frame to the first post-FRAP frame. The region whose intensity changed substantially was determined and was set as the FRAP region of interest (ROI). The intensity in this ROI was then determined for all frames and was corrected for background by subtracting the intensity in the noncellular areas (which was virtually equal to 0 arbitrary units, AU). To correct for bleaching induced during image acquisition, we produced a number of identical image series for each coverslip (typically 5), in which the exact same imaging procedure was followed, but without applying any laser intensity for the FRAP step. The decrease in fluorescence intensity during these series was measured, thereby providing the imaging-induced bleaching curve. The FRAP curves were corrected using the average bleaching curve for the respective protein (the bleaching curve was normalized by dividing it by the first point, and the FRAP curve was divided by the normalized bleaching curve). The FRAP curves were then fitted with single exponentials automatically, producing the results presented in Fig 2 and Appendix Fig S3. All curves were additionally plotted and were visually inspected by an experienced investigator, to avoid employing results from unusual or badly fitted curves.

### FCS experiments

For fluorescence correlation spectroscopy experiments (FCS), a home-built setup, integrated with an inverted microscope body (Olympus IX73, Olympus, Hamburg, Germany) was used. The experiments were performed using a diode-pumped laser with a wavelength of 491 nm (Cobolt Calypso Cobolt AB, Solna, Sweden). After exiting from the optical fiber, the laser light passed a clean-up filter (HC Laser Clean-Up MaxLine 491/1.9, AHF Analysentechnik, Tübingen, Germany) and was reflected by a dichroic mirror (Dual-Line zt488/532rpc, AHF Analysentechnik) into the microscope body. The light was focused onto the sample using a 60× water-immersion objective (UPlanApo, NA 1.2, Olympus). After passing an emission filter (488 LP Edge Basic Longpass filter, AHF Analysentechnik) and a pinhole (diameter 50 μm, Qioptiq Photonics Gmbh & Co. KG, Göttingen, Germany), the fluorescence light emitted by the sample was focused on two avalanche photodiodes (τ-SPAD, Picoquant GmbH, Berlin, Germany). The τ-SPADs were connected to a digital correlator card (ALV-7004 USB, ALV, Langen, Germany) used for autocorrelation measurements.

In order to identify measurement positions within the neuron, epifluorescence images were taken using a mercury arc lamp (X-Cite 120 PC Q, Excelitas Technologies, Uckfield, United Kingdom) as excitation light and a GFP fluorescence filter set (GFP HC BrightLine Basic Filter Set, AHF Analysentechnik). Images were acquired using a CCD-camera (Hamamatsu Orca R-2, Hamamatsu Photonics, Herrsching am Ammersee, Germany) controlled by Micro-Manager (Edelstein *et al*, 2010). An automated sample stage (Prior Scientific, Inc., Rockland, MA, USA) was used to access the different positions within the neurons. Before each experiment, a calibration measurement was performed using the fluorescent dye Atto488 (ATTO-TEC GmbH Siegen, Germany), which has a known diffusion coefficient

**Table 1. List of constructs, vectors, reference sequences, and linker sequences.**

| Protein of interest | Vector | RefSeq | Linker sequence |
|---|---|---|---|
| beta-Actin | pcDNA3.1 | NM_031144.3 | TGGGSGGGSGGGSAAA |
| alpha-SNAP | pcDNA3.1 | NM_080585.1 | TGGGSGGGSGGGSAAA |
| alpha-Synuclein | pEGFP-N1 | NM_001009158.3 | GTAGPGSIAT |
| Amphiphysin | pcDNA3.1 | NM_022217.1 | TGGGSGGGSGGGSAAA |
| AP180 | pcDNA3.1 | ×68877.1 | TGGGSGGGSGGGSAAA |
| AP2 alpha-C | pcDNA3.1 | ×53773.1 | TGGGSGGGSGGGSAAA |
| Calmodulin 1 | pcDNA3.1 | NM_031969.2 | TGGGSGGGSGGGSAAA |
| Clathrin light chain | pcDNA3.1 | NM_053835.1 | TGGGSGGGSGGGSAAA |
| Complexin 1 | pcDNA3.1 | U35098.1 | TGGGSGGGSGGGSAAA |
| Complexin 2 | pcDNA3.1 | NM_053878.1 | TGGGSGGGSGGGSAAA |
| CSP | pcDNA3.1 | NM_024161.2 | TGGGSGGGSGGGSAAA |
| Doc2a | pEGFP-N1 | NM_022937.2 | GSTVPSARDPPVAT |
| Dynamin 1 | pcDNA3.1 | NM_080689.4 | TGGGSGGGSGGGSAAA |
| EndophilinA1 | pcDNA3.1 | NM_053935.1 | TGGGSGGGSGGGSAAA |
| Epsin | pcDNA3.1 | NM_057136.1 | TGGGSGGGSGGGSAAA |
| Hsc70 | pcDNA3.1 | NM_024351.2 | TGGGSGGGSGGGSAAA |
| Intersectin 1-L | pcDNA3.1 | NM_001136096.1 | TGGGSGGGSGGGSAAA |
| Munc13 | pcDNA3.1 | NM_022861.1 | TGGGSGGGSGGGSAAA |
| Munc18 | pEGFP-N1 | L26087.1 | GSTPGG |
| NSF | pcDNA3.1 | NM_021748.1 | TGGGSGGGSGGGSAAA |
| PIP5KIgamma | pEGFP-C2 | NM_012398.2 | RPDSDLELKLRI |
| Rab3a | pcDNA3.1 | NM_013018.2 | TGGGSGGGSGGGSAAA |
| Rab5a | pEGFP-N1 | BC161848.1 | GSTPGG |
| Rab7a | pcDNA3.1 | NM_023950.3 | TGGGSGGGSGGGSAAA |
| SCAMP1 | pcDNA3.1 | NM_001100636.1 | TGGGSGGGSGGGSAAA |
| Septin 5 | pcDNA3.1 | NM_053931.4 | TGGGSGGGSGGGSAAA |
| SNAP23 | pcDNA3.1 | NM_022689.2 | TGGGSGGGSGGGSAAA |
| SNAP25 | pEGFP-N1 | NM_011428.3 | GSTPGG |
| SNAP29 | pcDNA3.1 | NM_011428.3 | TGGGSGGGSGGGSAAA |
| SV2B | pcDNA3.1 | AF372834.2 | TGGGSGGGSGGGSAAA |
| Synapsin 1 | pEGFP-N1 | NM_019133.2 | SGLRSREAAT |
| Synaptogyrin | pcDNA3.1 | NM_019166.2 | TGGGSGGGSGGGSAAA |
| Synaptophysin | pEGFP-N1 | NM_012664.3 | GSTPGG |
| Synaptotagmin 1 | pcDNA3.1 | NM_001033680.2 | TGGGSGGGSGGGSAAA |
| Synaptotagmin 7 | pcDNA3.1 | NM_021659.1 | TGGGSGGGSGGGSAAA |
| Syndapin 1 | pcDNA3.1 | NM_017294.1 | TGGGSGGGSGGGSAAA |
| Syntaxin1A | pEGFP-N1 | NM_053788.2 | LVSRARDPPVAT |
| Syntaxin 16 | pcDNA3.1 | NM_001108610.1 | TGGGSGGGSGGGSAAA |
| alpha-tubulin | pcDNA3.1 | NM_006082.2 | SGLRSR |
| VAMP1 | pcDNA3.1 | NM_013090.2 | TGGGSGGGSGGGSAAA |
| VAMP2 | pEGFP-N1 | NM_009497.3 | RILQSTVPRARDPPVAT |
| VAMP4 | pcDNA3.1 | NM_001108856.1 | TGGGSGGGSGGGSAAA |
| vATPase V0A1 | pcDNA3.1 | NM_031604.2 | TGGGSGGGSGGGnSAAA |
| vGluT1 | pcDNA3.1 | U07609.1 | TGGGSGGGSGGGSAAA |
| Vti1a-β | pEGFP-N1 | AF262222.1 | TGGGSGGGSAAA |

**Table 2.   List of primers used for molecular cloning.**

| Protein coded | Primers' sequences |
|---|---|
| Doc2a | 5′ GGTACCATCAGCTAGGGATCCACCGGTCGCCACC 3′<br>5′ CCTAGCTGATGGTACCGTCGACccGGCCAAC 3′ |
| Rab5 | 5′ GGATCTACACCTGGAGGAATGGTGAGCAAGGGCGAGGA 3′<br>5′ TCCTCCAGGTGTAGATCCGTTACTACAACACTGGCTTCTGGC 3′ |
| SNAP25 | Vector:<br>5′ TAATCTGCAGATTAATCTAGATAACTGATCATAATCAGCCATACCAC 3′<br>5′ GGTGGCTCGAGGCTAGCGGATCTGACGGTTCACTAAACCA 3′<br>mEGFP:<br>5′ CTAGCCTCGAGCCACCATGGTGAGCAAGGGCGAGG 3′<br>5′ TCCTCCAGGTGTAGATCCCTTGTACAGCTCGTCCATGCCG 3′<br>SNAP25:<br>5′ GGATCTACACCTGGAGGAATGGCCGAGGACGCAGAC 3′<br>5′ ATCTAGATTAATCTGCAGATTAACCACTTCCCAGCATCTTTGTTGC 3′ |
| Synaptophysin | 5′ GGATCTACACCTGGAGGAATGGTGAGCAAGGGCGAGGA 3′<br>5′ TCCTCCAGGTGTAGATCCCATCTGATTGGAGAAGGAGGTAGG 3′ |
| Vti1a-β | Vector-mEGFP:<br>5′ AGGACACACCGGCGGAGGAAGCGGC 3′<br>5′ CGGCCGCTTTACTTGTACAGCTCGTCCATGCCGTGAG 3′<br>Vti1a:<br>5′ CAAGTAAAGCGGCCGCGACTCTAGATCATAATCAGCC 3′<br>5′ GCCGGTGTGTCCTCTGACAAAAAAAGTGATGGCCGTCAG 3′ |

$(400 \ \mu m^2/s$ at 25°C) (Kapusta). The coverslips were mounted on the sample holder, in Tyrode's solution. For each protein analyzed, we acquired data from different positions in the axons of the neurons. Each position was measured at least 20 times, with acquisition times between 10 and 30 s for each round of acquisition. All data were then fitted with a Levenberg–Marquardt nonlinear least-square routine using a self-written Python code (Python Software Foundation, https://www.python.org/). Data in which large fluorescence peaks were observed, or with pronounced photobleaching, were excluded from the analysis.

**Organelle tracking analysis**

For organelle tracking, neurons were transfected following the lipofection procedure described above. Neurons were imaged 6–10 days after lipofection. The culture medium was replaced by pre-warmed Tyrode's solution (124 mM NaCl, 2.7 mM KCl, 10 mM $Na_2HPO_4$, 2 mM $KH_2PO_4$, pH 7.3). The temperature of the imaging chamber system was maintained at 37°C. Axons of transfected cells were then imaged using a Nikon Ti-E epifluorescence microscope (Nikon Corporation, Chiyoda, Tokyo, Japan) equipped with a $100 \times 1.4$ NA oil-immersion Plan Apochromat objective (Nikon Corporation, Chiyoda, Tokyo, Japan). Cells were imaged for up to 3 min with a picture taken every 500 ms. For a control experiment, cells were fixed with 4% PFA in PBS, quenched with 100 mM $NH_4Cl$ and imaged following the same procedure as done for live cell imaging.

The images were processed by a bandpass procedure (using a freely available MATLAB code, copyright 1997 by John C. Crocker and David G. Grier). This removes background and enables the observation of individual spots. This was followed by a particle finding and tracking routine (performed again using a freely available MATLAB code, copyright February 4, 2005, by Eric R. Dufresne, Yale University). The codes are available at: http://

site.physics.georgetown.edu/matlab/code.html. The particle tracks were then analyzed for speed.

**Overexpression analysis**

Overexpressing cells were immunostained for synaptophysin (as a synapse marker) and for the protein of interest (the overexpressed protein). The levels of the proteins of interest were measured by immunostaining in the transfected boutons, as well as in the non-transfected boutons. All analyzed boutons were selected by the synaptophysin immunostaining. They were then separated into overexpressing and non-overexpressing populations, based on the GFP signal. The overexpression levels were then derived by dividing the immunostaining intensity in the overexpressing boutons by that in the non-overexpressing boutons, both normalized to the respective synaptophysin levels, to account for differences in synapse size [see also (Truckenbrodt et al, 2018), for further examples of this procedure].

**Immunostainings and confocal and STED microscopy**

The cultures were immunostained for the proteins of interest, and for the synaptic vesicle marker synaptophysin (using a guinea pig antibody from Synaptic Systems; catalogue number 101 004) and the active zone marker bassoon (using antibodies from StressGene, catalogue number ADI-VAM-PS003-D, or from Synaptic Systems, catalogue number 141 002). The primary antibodies used for the proteins of interest are all noted in Appendix Fig S2. We used the following secondary antibodies: Cy2-conjugated goat anti-mouse or anti-rabbit antibodies for bassoon (Dianova); Cy3-conjugated goat anti-guinea pig antibodies for synaptophysin (Dianova); and Atto647N-conjugated goat anti-mouse or anti-rabbit antibodies for the proteins of interest (Synaptic Systems or Rockland). The cultures were fixed using 4% PFA in PBS for 45 min, were quenched using 100 mM

NH$_4$Cl in PBS for 15 min, and were then permeabilized using 0.1% Triton X-100 in PBS, in the presence of 1.5% BSA. The same medium was then used for incubations with primary and secondary antibodies. The samples were then washed several times with PBS and high-salt PBS (containing 500 mM NaCl) and were embedded in Mowiol or in 2,2′-thiodiethanol (TDE), as described (Wilhelm *et al*, 2014). The samples were imaged using a Leica TCS SP5 STED microscope, with a 100× oil-immersion objective (1.4 NA; HCX PL APO CS, Leica).

Confocal excitation was obtained with the 488-nm line of an Argon laser (green channel, for Cy2). Similarly, we used the 543-nm line of a Helium Neon laser for the orange (Cy3) channel. STED imaging was performed (for Atto647N) using a pulsed diode laser, at 635 nm, for the excitation channel, and a Spectra-Physics MaiTai tunable laser (Newport Spectra-Physics, Irvine, CA) for depletion at 750 nm. We set the AOTF filter of the system to the appropriate emission intervals for confocal imaging; an avalanche photodiode was used for the STED imaging. The AOTF filter of the microscope was used to select appropriate emission intervals for the different dyes. Signal detection was performed either by a photomultiplier (confocal mode) or by an avalanche photodiode (STED mode).

### Electron microscopy and reconstruction of synapses

Neuronal cultures were fixed and processed for electron microscopy as follows: Entire coverslips with neurons were immersed into 2.5% glutaraldehyde buffered with 150 mM sodium cacodylate containing 2 mM CaCl$_2$ at pH 7.4 for 30 min. Samples were washed with sodium cacodylate and postfixed with 1% osmium tetroxide in 150 mM sodium cacodylate and 1.5% potassium ferrocyanide at room temperature for 30 min. After washing, samples were again postfixed in 1% osmium tetroxide in sodium cacodylate for another 30 min. Samples were thoroughly washed in distilled water and block contrasted in 2% uranyl acetate at room temperature for 60 min. Next, specimens were *en bloc* contrasted with lead (33 mg lead nitrate in 10 ml of 30 mM aspartic acid; modification of (Walton, 1979) at 60°C for 60 min. After washing in water, specimens were dehydrated in an ascending acetone series (50, 70, and 90%) followed by two steps in dry 100% ethanol for 20 min each. A mixture of Epon and Spurr's resin (1 + 1) was used to infiltrate the cells, and blocks were polymerized at 60°C for 2 days. Cultures were then thin-sectioned in the plane of the coverslips with a UC6 ultramicrotome (Leica Microsystems, Wetzlar, Germany). Serial sections were imaged with a JEOL 100CX microscope equipped with a mid-mount 4MP Hamamatsu camera controlled by AMT software (Advanced Microscopy Techniques, Woburn, MA).

For synapse reconstructions, we relied on serial sections which we aligned manually in Photoshop (Adobe Systems, San Jose, CA). The different elements of the synapse (active zone, membrane, vacuoles, mitochondria, and vesicles) were marked manually on each consecutive frame, and the values were compiled using a self-written MATLAB routine, as described (Rizzoli & Betz, 2004; Denker *et al*, 2011b).

### Modeling analysis I: Generation of the 3D synapse

We first determined different parameters from 30 different 3D synapse models. We then selected one synapse whose various parameters (active zone surface, volume, vacuole volume and number, synaptic vesicle volume and number, mitochondria volume) were close to the average of all synapses (difference from the average of all 30 synapses of only 26.9%, over all measured parameters). As the 3D measurements were performed from chemically fixed samples, we accounted for their shrinkage during fixation and plastic-embedding (Gaffield *et al*, 2006).

To obtain a widely applicable model, we then prolonged the axonal connections of this synapse for several micrometers, as shown in the 3D view of the model (Appendix Fig S17), relying on published super-resolution axon diameter and shape measurements (Xu *et al*, 2013). The model, initially obtained at an approximately 3 nm X-Y plane resolution, and a 70 nm axial resolution, was then modified to obtain 25 × 25 × 25 nm voxels, as presented in Appendix Fig S17, in which the different organelles were placed according to the measured positions from the original EM images. The 25 × 25 nm size was chosen as this is also the value used in the live FRAP movies. Synaptic vesicles were placed in multiple pixels, since the vesicle volume [for a 42 nm diameter (Takamori *et al*, 2006)] is approximately 2.5-fold larger than the voxel volume. As we planned to model protein binding on vesicles and/or on components of the vesicle cluster, we increased the number of voxels allotted to synaptic vesicles approximately 2-fold, to account for large molecules being able to bind vesicles from a distance of a few nanometers, or for molecules binding other elements that are themselves bound to vesicles, but not directly to vesicles (as, e.g., would be the case for actin, which is likely to bind to actin strands or to synapsin in the vesicle cluster, but probably not to the vesicle surfaces).

### Modeling analysis II: moving particles in the 3D synapse

We placed particles in the 3D synapse model and allowed them to move randomly. The particles either moved only in the plasma membrane, or in the cytosol. The movement in the plasma membrane was modeled in a simple fashion, with the particles always moving at a specified single-step velocity. This approximated a Gaussian distribution, with a width of ~25 nm; the following displacements were modeled: 25, 50, 75, 100, 125, 150, and 200 nm as well as several intermediate displacements.

The same type of motion was also induced in the cytosol. In addition, the particles that reached synaptic vesicle pixels also remained bound to them for defined time periods, ranging from 1 movement steps (no binding to vesicles) to 200 movement steps. Up to 1000 particle tracks were generated for every movement model.

### Modeling analysis III: placing fluorophores on the particle tracks

To use them for further analysis, the movement tracks for one model (for every single-step velocity and for every vesicle binding capacity) were combined, and fluorophores were placed on their various particle positions. For this, we convoluted the particle positions with a confocal point-spread-function (PSF), measured exactly as in the actual FRAP movies. The actual PSF is shown in Fig 6. This results in fluorescence movies for the different movement models, in which the fluorophores corresponding to the different particle positions overlap, creating a realistic synapse image, as shown in Fig 6.

## Modeling analysis IV: generating *in silico* FRAP movies

To mimic FRAP situations, we eliminated (bleached) the fluorophores from regions identical in size to those observed in the actual FRAP movies, and we then allowed the particles to move, to induce the signal recovery. The particles continued to move along their tracks, and some that were not bleached entered the FRAP zone, generating a recovery signal. To account for the diffusion of particles to and from neighboring axonal regions, we forced particles that exited on either open side of the axon to return through the opposite side, thereby generating an equilibrium movement situation. Bleached particles that exited the axon became fluorescent again upon re-entry, to mimic the exchange of such particles for fluorescent ones from the neighboring areas, which is known to happen *in vivo* (Darcy *et al*, 2006).

We generated, in independent *in silico* experiments, FRAP data for both the synapse and for the axon, by choosing appropriate FRAP regions in the 3D model.

## Modeling analysis V: the analysis of the *in silico* FRAP movies

The resulting artificial FRAP movies were fitted with single exponential curves exactly as performed for the biological FRAP movies. As for each model, we generated both axon and synapse FRAP data, the fits resulted in two parameters: (i) Synapse FRAP τ (time constant) values and (ii) Axon FRAP τ (time constant) values.

For the membrane movement models, the time constants for the axonal and synapse FRAP correlated linearly to each other, and occupied a line that overlapped broadly with the measured values for the proteins that we expect to diffuse as single molecules in the plasma membrane. The real data were best reproduced when length of the model time step (the interval between movement steps) was chosen as 5.7 ms.

For the cytosolic proteins, we obtained the synapse and axon FRAP time constants for different single-step velocities (from 25 to 250 nm), combined with different periods of binding synaptic vesicles (from 1 movement step to 200 movement steps). Placing a 3 ms value for the length of the model time step resulted in a good correlation between the models and all of the measured data.

## Modeling analysis VI: determining the models that best fit the real data

Each real experiment provides two values: the synapse FRAP τ and the axon FRAP τ. Similarly, each model also provides a synapse FRAP τ and an axon FRAP τ. One can thus calculate the two-dimensional distance between each experimental pair of values, and each model pair of values. For each protein, we determined the 3–5 models (with different single-step velocities and/or different vesicle binding) that were closest to the respective experimental synapse and axon τ values. The diffusion coefficients reported for the proteins correspond to the single closest model, with the errors corresponding to the range of values of the 3-5 models.

## Modeling analysis VII: determining the diffusion coefficients

We then determined the diffusion coefficients of the different models by determining their mean square displacements (Qian *et al*,

1991), relying on the Einstein–Smoluchowski equation (Islam, 2004). The diffusion coefficients were determined at different positions within the synapse or in the axon, and, for the soluble proteins, also in the vesicle cluster.

## Modeling analysis VIII: the case of bona fide synaptic vesicle or endosome proteins

For proteins found mainly in synaptic vesicles or in endosomes, the procedure outlined above cannot reproduce the synapse time constants (for proteins found mainly in the synapse) or the axonal tau constants (for VAMP4, which tends to be found in axonal endosomes). For these proteins, we extracted the model diffusion values that fitted the movement of the molecules in the compartment in which they would move as single proteins (in the axon, for most of these proteins, or in the synapse, for VAMP4). We then extrapolated the diffusion of the free molecules in the other compartment (synapse or axon, respectively) from the models. For proteins such as synaptogyrin, vGlut, and SCAMP, which have only been measured in the synapse, we assigned diffusion coefficients by analogy to the most similar vesicle molecules (synaptophysin for the first two, vATPase and VAMP1 for SCAMP).

To validate this procedure, we determined the average diffusion coefficient of the synaptic vesicles. The procedure explained above provides the diffusion coefficient of free molecules in the synapse ($D_{free}$). Applying the diffusion models to the time constant obtained in real experiments in the synapses provides a diffusion coefficient for a mixture of molecules found in vesicles and free molecules ($D_{free + vesicles}$). At the same time, the proportion of molecules found in vesicles is identical to the fraction of immobile molecules in the synapse, since very little vesicle exchange can take place during our short FRAP procedure. Thus, knowing the two diffusion coefficients, as well as the proportions of the molecules involved, one can extract the diffusion coefficient for the vesicle-bound proteins. The average value obtained over all vesicle proteins is ~0.0102 μm$^2$/s, similar to what has been reported in the literature (see main text).

## Analyzing the proportion of molecules in the axon or in the synapse

To determine the proportion of molecules in the axon and in the synapse (Fig 7A–C), we turned to the *in silico* FRAP movies, and simply counted particles found in the axon or in the synapse at the FRAP time point. The respective values were compared to results obtained by measuring the intensity in manually measured ROIs in synapses and in neighboring areas, from immunostaining experiments (the synapse positions were determined by immunostaining for synaptophysin and the active zone protein bassoon, as indicated in the Materials and Methods section on immunostaining procedures).

## Generating and analyzing artificial STED images

To obtain and analyze artificial STED images, we first noted the number of antibodies that would be expected to bind within one synapse, for each protein (Richter *et al*, 2018). We then selected randomly a number of protein positions (taken from the positions

behind the artificial FRAP movies) corresponding to the number of antibodies, and convoluted these positions with a measured STED point-spread-function, measured exactly as in the STED immunostainings for the respective proteins. The remaining pixels in the resulting 2D images were covered with a measured STED background, containing measured salt-and-pepper-like noise. 100 simulated STED images were generated for every protein, and the STED spots were automatically detected, by first filtering the images using a bandpass filter, and then selecting as regions-of-interest all spots that were above a user-defined threshold (same for all images, for all proteins; chosen so as to eliminate salt-and-pepper noise). The full width at half maximum (FWHM) and the intensity of the spots were then determined from Lorentzian fits to the spots (Willig *et al*, 2006; Maidorn *et al*, 2018). Real STED immunostaining images were then analyzed in a similar fashion. The regions of interest were first selected in the synaptophysin channel, to restrict the analysis to synapses. This type of analysis has also been recently discussed in (Maidorn *et al*, 2018). For display purposes, we deconvolved the STED images using Huygens Essential software (Scientific Volume Imaging, Hilversum, The Netherlands), based on the inbuilt routines generated by the company.

### Construction of the dynamic graphical model

The individual protein views were constructed using custom-written plug-ins and scripts in the 3D software Autodesk Maya (Autodesk Inc., San Rafael, CA). Protein structure information was derived from the UniProt database. The same individual protein view models were used as in Wilhelm *et al* (2014), and the references used are presented in the particular paper. When available, we used protein database (PDB) coordinates in order to reconstruct proteins. If not available, we relied on structure information provided by a number of prediction servers. We used the following types of information: secondary structure information (http://bioinf.cs.ucl.ac.uk/psipred/); disorder calculations (http://mbs.cbrc.jp/poodle/poodles.html; http://mbs.cbrc.jp/poodle/poodle-w.html; alignment (http://web.expasy.org/sim/); predictions of coiled coil regions (http://toolkit.tuebingen.mpg.de/pcoils; http://mbs.cbrc.jp/poodle/poodle-l.html), information on transmembrane domains (http://www.ch.embnet.org/software/TMPRED_form.html); information on glycosylation domains (http://www.glycosciences.de/modeling/glyprot/php/main.php); domain identification (http://smart.embl-heidelberg.de/index2.cgi); and the presence of homologue proteins (http://web.expasy.org/blast/).

As mentioned in the main text, simulated different particle motion behaviors, with different movement speeds, in order to find the behaviors that most closely reproduced the FRAP results. We transformed the particle motion in artificial FRAP movies by applying fluorophore point-spread-functions onto the tracks. We then compared the results to the original FRAP data, in order to find the models that best reproduced the biological time constants in the axons and in the synapses. We then placed the protein structures in the 3D space of the model synapse, relying on the same movement tracks we used in the rest of this work. For each protein type, we used for the graphical models a number of protein tracks from the model that had reproduced best the FRAP behavior of the respective protein. The number of tracks was chosen as equal to the expected protein copy number for the respective protein (Richter *et al*, 2018).

The protein views were then placed on every pixel of the tracks and were also allowed to turn around their own axis. Synaptic vesicles were presented according to their known composition (Takamori *et al*, 2006), relying on previously generated models (Wilhelm *et al*, 2014). The vesicles are typically shown in grayscale. To avoid confusing the viewer, no vesicle motion is shown (or only a rotational/vibrational motion); this is a reasonable procedure, since the net diffusive vesicle motion is expected to be extremely limited for the time interval we show.

### Statistical analysis

All FRAP data are presented in detail in the extensive Appendix Fig S3. Other figures may also present data subsets as means ± SEM, or means ± SD, as indicated in the respective figure legends. All statistical comparisons are presented in detail in the respective figure legends. For comparisons of synaptic datasets, we relied on Kruskal–Wallis tests, followed by Mann–Whitney comparisons, and, where necessary, additional corrections for multiple testing, using the Bonferroni or Benjamini-Hochberg procedures, as indicated in the respective figure legends.

# Code and materials availability

All routines are available upon request to S.O.R., and all requests for materials and correspondence should be directed to S.O.R. (srizzol@gwdg.de).

**Expanded View** for this article is available online.

### Acknowledgements

We thank Reinhard Jahn (Max Planck Institute for Biophysical Chemistry, Göttingen, Germany) for helpful comments on the manuscript. We thank Oleh Rymarenko (Max Planck Institute for Biophysical Chemistry, Göttingen, Germany) for helping with data plotting automation. We thank Reinhard Jahn (Max Planck Institute for Biophysical Chemistry, Göttingen, Germany), Flavia Valtorta (San Raffaele Vita-Salute University, Milan, Italy), and Tiago F. Outeiro (University Medical Center Göttingen, Germany) for providing several plasmids used in this work. The work was supported by grants to S.O.R. from the European Research Council (ERC-2013-CoG NeuroMolAnatomy) and from the Deutsche Forschungsgemeinschaft (SFB1190/P09, SFB 1286/B02). We acknowledge the support of project SFB 1286/Z02, led by Stefan Bonn (University Hospital Hamburg-Eppendorf (UKE), Germany). Also supported by the DFG under Germany's Excellence Strategy—EXC 2067/1- 390729940.

### Author contributions

SR performed all fluorescence imaging experiments. J-EU and TS performed the electron microscopy experiments. EP and SR performed and analyzed the FCS measurements under the supervision of SK. EFF participated in the refinement of experimental conditions. SR, ST, and SOR analyzed the data. BR and SOR produced the 3D visualization. SR and SOR wrote the initial draft of the manuscript, which was then refined by all other authors.

### Conflict of interest

The authors declare that they have no conflict of interest.

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
