## [Review Process File · The EMBO Journal]

A comparative analysis of the mobility of 45 proteins in the synaptic bouton

Sofia Reshetniak, Jan-Eike Ußling, Eleonora Perego, Burkhard Rammner, Thomas Schikorski, Eugenio Fornasiero, Sven Truckenbrodt, Sarah Köster, and Silvio Rizzoli

DOI: [10.15252/embj.2020104596](https://doi.org/10.15252/embj.2020104596)

Corresponding author(s): Silvio Rizzoli (srizzol@gwdg.de)

Review Timeline:

Submission Date:	31st Jan 20
Correspondence	19th Mar 20
Editorial Decision:	23rd Mar 20
Revision Received:	7th Apr 20
Editorial Decision:	7th May 20
Revision Received:	20th May 20
Accepted:	29th May 20

Editor: Karin Dumstrei

Transaction Report:

Dear Silvio,

Thank you for submitting your manuscript to us. I am sorry for the slight delay in getting back to you with a decision but it is a very busy time.

Your study has now been seen by three referees and their comments are provided below. As you can see the response from the referees is a bit mixed. The main concern raised by all referees is that the analysis relies too much on over expression analysis and that we would need to some further data to support that the same behavior seen is also seen for endogenous proteins - not for all of them but for some. I don't know what experiments you have on hand or what you could add in a revised manuscript. I therefore find it best to do a pre-consultation upfront to see what you can add in a revised version. Based upon your point-by-point response I will then take the decision on the study.

Just contact me if you have any further questions

With best wishes

Karin

Referee #1

As correctly pointed out in the introduction of this manuscript, molecular mobility of synaptic proteins has hardly been investigated, at least not in a systematic comparative manner. In the submitted work, the mobility of a spectrum of synaptic proteins is systematically estimated using FRAP analysis of XFP labelled proteins. Concretely, they measured 47 proteins, including controls such as free cytosolic GFP, or membrane-bound GFP. They stick to overexpression of GFP-tagged variants of the proteins, for the reason that it is/was the "only efficient solution when large numbers of constructs need to be analysed". They then run their data set through a spectrum of correlations with protein structural and cell biological entities.

Judged from both a biological and biochemical perspective, the idea of retrieving systematic data concerning the mobility of synaptic proteins within synaptic boutons (and axons in comparison) is appealing. This also as these data might per se provide a basis and resource for modelling. This said, though I do to a degree understand their decision to work with cDNA overexpression constructs, concerns remain regarding the in vivo relevance of the data. I do see their arguments that the level of overexpression is moderate. Still, the sheer presence of the unlabelled endogenous protein might just significantly change the behaviour of the overexpressed species. In my eyes, and it might be a wrong impression, there is surprisingly little difference they report between different classes of SV and active zone proteins, though I would have expected this given the little bit of e.g. Unc13 being expressed from endogenous locus. Obviously, expressing from the endogenous locus in cultivated neurons is no longer out of reach given the availability of CRISPR/Cas-technology.

Taken together, I respect their substantial effort associated with this work, which I think per se is well executed and described. However, I am afraid my concerns preclude publication of the manuscript in EMBO J, at least in its current form.

Could they at least for a few selected cases:

- Compare FRAP behaviour in presence/absence of the respective endogenous protein?
- On-locus tag and compare results to the cDNA overexpressed situation?

In case of a favourable outcome, this might change the situation.

Referee #2

To depict the protein dynamics of pre-synapses, the authors measured the apparent mobility of 45 GFP-tagged protein species in the pre-synapses and axons of cultured neurons using FRAP. These mobility measures correlate with protein interactions with synaptic vesicles, lifetime, amino-acid content, and even mRNA composition. The authors advocate for the use of such

large mobility datasets in modeling molecular kinetics of presynaptic processes such as exo- and endocytosis.

The investigation of 45 protein species by microscopy is impressive. The dataset and its correlative studies are potentially informative though the control experiments could be improved. Main comments are expanded below:

- 1) In the material and methods, a large developmental range (DIV9-14) were used in the experiments. Why is such a variable range of timepoints used? These timepoints span from before and during synaptogenesis and the initial stages of synaptic maturation. These different developmental stages might change the interpretation of the results, as protein dynamics likely change in developing vs immature vs mature presynaptic boutons. Additionally, why are two different types of transfection reagents/protocols used in the same experiments?
- 2) The authors stressed at multiple points within the manuscript that their results are not due to overexpression artefacts. The data presented in figure 3 are not convincing. In overexpressing a tagged protein of interest, staining for "endogenous protein" will also detect the tagged population. In the example shown for NSF, the green signal, representing the fusion protein, should account for more than half of the total NSF signal- indicated in blue- this is not the case. The authors themselves acknowledged that previous work on a Munc13 knock-in demonstrated far slower mobility (minutes) as opposed to theirs (seconds). They imply that this may be the only one suffering from such issues- but why do they make this conclusion?
- 3) Fig. S12 has shown that in the synapse, immobile fraction and mobility time constants correlate. Does this mean mobility simply reflects protein interaction with presynaptic machinery? Fig. 7 shows that vesicle enrichment results in slower synaptic mobility. If so, is immobile fraction not a more intuitive measure of a protein's involvement in presynaptic dynamics? What more does movement rate of a mobile species tell us? The authors should explain or demonstrate.
- 4) Why does this correlation between immobile fraction and movement rate break down in the axons (Fig. S12)?
- 5) The authors mentioned criticism of quantum dot tracking to justify the use of FRAP. The authors should briefly summarize the criticism cited in Lee et al., 2017- and discuss the relative bulk added by the GFP molecule as opposed to a quantum dot of different sizes.
- 6) Fig. S1 shows consistently lower synaptic fractions in GFP-tagged experiments compared with immunostaining. Why? How does this affect subsequent data interpretation?
- 7) The authors test the role for active transport by assessing the dynamics of organelles during their imaging window. The paragraph describing figure 5 does not really describe the figure or the rationale for the experiment clearly. A more direct assessment would be to depolymerize microtubules or actin immediately prior to bleaching- allowing them to extract any role of active transport from the recovery curve itself. An example of a non-organelle associated protein should also be shown in the figure.
- 8) In Fig. 6, could the synaptic species be less mobile simply because of the geometry, as mentioned in the Discussion? How does one distinguish this effect from vesicle enrichment (Fig. 7)?
- 9) Fig. 7C left correlation looks suspicious. How much does upper right datapoint influence the correlation? Why is it such an outlier?
- 10) It is not clear why the authors thought to look for correlations at the mRNA level. It is also not clear if the authors are comparing the endogenous mRNA or their FRAP reporter mRNA. The authors should elaborate.

Referee # 3

Summary of the major findings: In this manuscript the authors use FRAP to infer the mobility of 45 GFP-tagged proteins in cultured hippocampal neuron axons and synapses.

Major shortcomings:

1. The study appears unfocused. I had trouble pinpointing the precise question that this study was trying to address. Further there is insufficient conceptual advancement in this study and a lack of physiological relevance.
2. They claim previous studies tracking mobility of these proteins did not enable analysis of protein structure in relation to synaptic mobility. Using FRAP does not enable protein structure analysis and FRAP does not provide molecular trajectory analysis. There are other techniques that can be utilized to assess this, which have already been carried out in the literature. Additionally, the authors suggest they are generating a protein mobility database for future modeling analysis and that they look at only basal culture conditions as network stimulation would not be physiologically relevant. However, these results are not the definitive protein mobilities, but ensemble measurements specific only to GFP-overexpressed proteins in hippocampal neuron cultures. This data therefore falls short of their aim of generating a protein mobility database, and does not provide any further biological insights for the individual proteins of interest as many have already been studied which they themselves state.
3. Figure 1 should provide references for each protein to accompany the validation score.
4. Figure 2a does not sufficiently show that tagged proteins localize to synaptic and axonal areas. Several labels should be used to identify these compartments, and the localization of GFP in relation to these structures.
5. I have serious concerns over the over-expression analysis. They quote in the introduction they only investigated neurons with mild expression of the GFP-tagged constructs and in the figure 3 legend with moderate expression levels. What do they suggest is mild/moderate? What is the exact threshold value used to determine whether the neuron could be analysed? In the materials and methods for the overexpression analysis it is not clear what they define as 'non-overexpressing boutons'. Is this an Immunolabelling of endogenous (non-infected) neurons to establish the threshold? Or have they assumed a threshold for overexpression within the infected neuron population? In which case how have they come to this threshold value to separate these 2 populations?
6. In figure 3 they state the error bar indicates the 75th percentile. This suggests the overexpression levels have a greater range than those visible in figure 3, and thus the overexpression levels do not just range from 1.2-2 fold as stated in the introduction.
7. I would suggest they consult a statistician. The use of the Benjamini-Hochberg procedure in figure 4 is not fully justified, as these are strictly only bivariate comparisons of the overexpression level vs mobility in each individual protein. The n number is also relatively low and thus the use of the Benjamini-Hochberg procedure can easily induce false negatives in such small populations. Multiple tests are not being carried out on the same protein, and each protein is independent of the others, thus they are not correcting for multiple testing.
8. It would be more helpful to see the linear regression plotted on each of the graphs shown in figure 4.
9. What is the spot size of the FRAP area? And what is the area being used to analyse recovery rates? These values will have a huge influence on the time constants.
10. The tracking analysis of organelle-bound proteins cannot be extrapolated to deduce whether active organelle processes are affecting the FRAP estimates for all the analysed proteins.
11. The materials and methods section on the organelle tracking analysis is not detailed enough to understand how the images were acquired or how the analysis was carried out.
12. The authors make an assumption that the mobility they observe accounts for protein-specific interactions and that this confirms their expression conditions, yet the data does not directly support this claim. The wording is very misleading as they also state they compare FRAP time constraints with strength of association of each protein to synaptic vesicles, yet they do not measure association strength.
13. The authors state the fold difference between synaptic and axonal time constants is similar to the fold difference between membrane and soluble proteins and that this suggests the synaptic environment has a similar influence on the movement of both types of proteins. Only absolute

values of time constants could be used to deduce this, not fold changes.

14. In the discussion the authors compare FRAP only to FCS and not single particle tracking methods and thus a global view of the literature is missing.

15. Figure S12 - why do they have negative immobile fractions?

16. Grammatical errors and misleading text throughout.

Dear Silvio,

Thank you for the constructive talk regarding what data you can add to address the concerns raised by the referees. I appreciate the discussed data on how to further validate the dataset and would like to invite you to submit a suitable revised manuscript that addresses the raised concerns as discussed. The revision link is provided below.

Let me know if there is anything further to discuss

With best wishes

Karin

Karin Dumstrei, PhD
Senior Editor
The EMBO Journal

The revision must be submitted online within 90 days; please click on the link below to submit the revision online before 21st Jun 2020.

Replies to the Referee comments. The comments are shown in italics, with our replies in regular font.

Referee #1

As correctly pointed out in the introduction of this manuscript, molecular mobility of synaptic proteins has hardly been investigated, at least not in a systematic comparative manner. In the submitted work, the mobility of a spectrum of synaptic proteins is systematically estimated using FRAP analysis of XFP labelled proteins. Concretely, they measured 47 proteins, including controls such as free cytosolic GFP, or membrane-bound GFP. They stick to overexpression of GFP-tagged variants of the proteins, for the reason that it is/was the "only efficient solution when large numbers of constructs need to be analysed". They then run their data set through a spectrum of correlations with protein structural and cell biological entities.

Judged from both a biological and biochemical perspective, the idea of retrieving systematic data concerning the mobility of synaptic proteins within synaptic boutons (and axons in comparison) is appealing. This also as these data might per se provide a basis and resource for modelling.

We thank the referee for the comments.

*This said, though I do to a degree understand their decision to work with cDNA overexpression constructs, concerns remain regarding the in vivo relevance of the data. I do see their arguments that the level of overexpression is moderate. Still, the sheer presence of the unlabelled endogenous protein might just significantly change the behaviour of the overexpressed species. In my eyes, and it might be a wrong impression, there is surprisingly little difference they report between different classes of SV and active zone proteins, though I would have expected this given the little bit of e.g. *Unc13* being expressed from endogenous locus. Obviously, expressing from the endogenous locus in cultivated neurons is no longer out of reach given the availability of CRISPR/Cas-technology.*

*Taken together, I respect their substantial effort associated with this work, which I think per se is well executed and described. However, I am afraid my concerns preclude publication of the manuscript in *EMBO J*, at least in its current form.*

Could they at least for a few selected cases:

- Compare FRAP behaviour in presence/absence of the respective endogenous protein?*
- On-locus tag and compare results to the cDNA overexpressed situation?*

In case of a favourable outcome, this might change the situation.

We thank the referee again for the comments. We have now performed the following controls:

- 1) We compared the FRAP curves for GFP-tagged synaptotagmin 1 with FRAP curves obtained after labeling native (endogenous) synaptotagmin 1 with an antibody that binds the intravesicular (surface-exposed) domain of this molecule (published in our own work; (Kamin et al., 2010)). The curves were identical. Please see the new Figure S6, panel a.
- 2) We compared the FRAP curves for GFP-tagged VGLUT1 with FRAP curves obtained in cultured mouse hippocampal neurons from knock-in Venus-tagged VGLUT1 (Herzog et al., 2011). The curves were again identical. Please see the new Figure S6, panel b.
- 3) We performed a similar comparison for GFP-tagged alpha-synuclein, with results from knock-in Syn-GFP mice (under minimal expression, mimicking human disease (Spinelli et al., 2014)). The estimated FRAP values, calculated according to the respective

publication, are very close to the results we obtained. Please see the new Figure S6, panel c.

- 4) We compared the FRAP curves of the proteins that are known to be exceptionally enriched in synaptic vesicles, and are not present at substantial levels in any other synaptic compartment, to FRAP curves of synaptic vesicles, obtained after labeling the vesicles with an FM dye (Shtrahman et al., 2005). The FRAP curves of these proteins (synaptophysin, synaptogyrin, VGLUT1 and SV2) are virtually identical to the FRAP curves of vesicles labeled with FM 1-43. Please see the new Figure S6, panel d. Also, please note that these proteins show a slower and more limited recovery than synaptotagmin 1, which is known to be present at higher levels on the plasma membrane, and must therefore be more mobile.

Our results are also very similar to previous FRAP measurements for actin and calmodulin, which have been investigated in a variety of cDNA overexpression studies, mostly in dendrites (for example (van Bommel et al., 2019; Koskinen and Hotulainen, 2014; Koskinen et al., 2012; Petersen and Gerges, 2015)). It is unlikely that the overexpression of GFP-actin or GFP-calmodulin affected strongly the dynamics of these key neuronal proteins, as this would have damaged the respective neurons. No such damage was observed in the studies mentioned. This would enable us to conclude that in the respective studies the biological effects of overexpression were negligible. In turn, this implies that the effects of overexpression for actin and calmodulin are probably also negligible in our work.

Overall, the only measurement from the literature that we could not reproduce is that of Munc13-YFP (Kalla et al., 2006), as already explained in our original manuscript. A substantial fraction of the difference to the Kalla et al. results may be due to the way the FRAP experiments were performed and analyzed, as we mentioned in the original manuscript.

In principle, we could also follow the referee's suggestion to express proteins from the endogenous locus in primary rat cultured neurons. However, this procedure is still very difficult. Moreover, substantial experimental work is at the moment is unfortunately impossible, due to the coronavirus outbreak, which forced the shutdown of multiple laboratories. With a fully functional laboratory, we could follow this approach. In the current situation, however, access to critical resources, including Cas9-expressing rats, is unavailable due to the shutdown of the respective facilities, which is likely to be prolonged for the foreseeable future.

In the absence of such experiments, we decided to address further the comment of the referee, by approaching the FRAP issue from a different perspective. We have analyzed the FRAP results thoroughly, and we have obtained realistic movement parameters from the data, in the form of diffusion coefficients. The modeling analysis is explained in the revised manuscript (please see the new Figure 5, and the respective paragraphs in the Methods section). The movement parameters we obtained could then be compared to any dataset desired from the literature or from our own experiments, to determine whether the results are substantially corrupted by the technology we used.

From our analysis of the FRAP data we generated the following validations, in addition to the list presented above:

- 1) Our FRAP-derived data reproduced the diffusion coefficients of mEGFP or syntaxin 1, as previously presented in the literature (please see the new Figure 7).
- 2) The FRAP-derived data reproduced the diffusion coefficient of synaptotagmin 1, previously obtained in our own live STED single-molecule tracking experiments, obtained

- by tagging the endogenous protein (Kamin et al., 2010; Westphal et al., 2008); please see the new Figure 7).
- 3) From the behavior of synaptic vesicle proteins we could calculate the diffusion coefficient of synaptic vesicles, which again reproduced very well the average value known from the literature (please see the new Figure 8).
 - 4) Our results confirmed the expectation that “free” GFP (not linked to any other protein) is the only soluble protein able to diffuse in the synapse without binding to synaptic vesicles (please see the new Figure 6).
 - 5) The data obtained enabled us to reproduce the differential distribution of the proteins in the axon and in the synapse, with determination coefficient (R^2) of 0.7-0.8, when compared to immunostainings (please see the new Figure 7a-c).
 - 6) We also used these data to determine the copy number of proteins per synaptic vesicle, for the proteins that are known to reside in vesicles. This reproduced well the copy numbers measured in the past by protein biochemistry (please see the new Figure S18).
 - 7) We used the overall protein positions provided by the analysis to reconstruct stimulated emission depletion (STED) images for the different proteins. The model images reproduced well the spot size and spot intensity distributions from the real images, from STED analyses of immunostained preparations (R^2 of 0.7-0.8; please see the new Figure 7d-i).
 - 8) The protein positions measured by our analysis were able to predict the fraction of membrane proteins found in the synapse, which correlated well with the enrichment of these proteins on the synaptic vesicles, known from biochemical experiments in the literature (Takamori et al., 2006). Please see the new Figure S19.
 - 9) Similarly, our analysis of FRAP parameters could predict the enrichment of the soluble proteins in the synaptic vesicle cluster. This correlated well with the enrichment of soluble proteins on purified synaptic vesicles (Takamori et al., 2006). Please see the new Figure S20.

We hope that these validations are sufficient to indicate that our work is not strongly biased by the overexpression approach we used.

Referee #2

To depict the protein dynamics of pre-synapses, the authors measured the apparent mobility of 45 GFP-tagged protein species in the pre-synapses and axons of cultured neurons using FRAP. These mobility measures correlate with protein interactions with synaptic vesicles, lifetime, amino-acid content, and even mRNA composition. The authors advocate for the use of such large mobility datasets in modeling molecular kinetics of presynaptic processes such as exo- and endocytosis.

The investigation of 45 protein species by microscopy is impressive. The dataset and its correlative studies are potentially informative though the control experiments could be improved.

We thank the referee for the comments. Please also refer to our replies to Referee 1, which explain 13 additional controls we have performed, and which we are now including in the manuscript.

Main comments are expanded below:

1) *In the material and methods, a large developmental range (DIV9-14) were used in the experiments. Why is such a variable range of timepoints used? These timepoints span from before and during synaptogenesis and the initial stages of synaptic maturation. These different developmental stages might change the interpretation of the results, as protein dynamics likely change in developing vs immature vs mature presynaptic boutons.*

We apologize for the confusion, which comes from the description of our transfection procedure, which is relatively long and complex. Typically cultures were used from DIV 14 onwards. No cultures younger than DIV 11 were used.

Additionally, why are two different types of transfection reagents/protocols used in the same experiments?

The laboratory employed a relatively inefficient calcium phosphate transfection protocol at the start of the project. A more efficient lipofection procedure was established during the project, which enables the transfection of more neurons per coverslip. The large majority of the proteins have been studied using the lipofection procedure.

We observed no differences between the FRAP curves obtained using different transfection protocols for individual proteins. We could show this in the form of a supplementary figure, if the Referee feels it would be important to showcase this issue.

2) *The authors stressed at multiple points within the manuscript that their results are not due to overexpression artefacts. The data presented in figure 3 are not convincing. In overexpressing a tagged protein of interest, staining for "endogenous protein" will also detect the tagged population. In the example shown for NSF, the green signal, representing the fusion protein, should account for more than half of the total NSF signal- indicated in blue- this is not the case.*

We are not sure what the referee refers to, since one cannot compare visually the intensity of the different color channels to derive a quantitative impression. We analyzed the respective images thoroughly, as indicated in the figure. As NSF is a very abundant protein, and as we consistently tried to only analyze proteins with moderate expression levels, it was not surprising to note that the overexpression level was ~108% in the respective boutons, which is relatively low.

The authors themselves acknowledged that previous work on a Munc13 knock-in demonstrated far slower mobility (minutes) as opposed to theirs (seconds). They imply that this may be the only one suffering from such issues- but why do they make this conclusion?

We apologize for the impression this paragraph made. We simply wanted to point out and discuss one known case in which knock-in measurements have been performed, and their results were substantially different from our data.

We compared our data to all other such works that we knew of, and the respective comparisons showed no disparities between our work and experiments on knock-in proteins or native proteins:

- 1) We compared the FRAP curves for GFP-tagged synaptotagmin 1 with FRAP curves obtained after labeling native (endogenous) synaptotagmin 1 with an antibody that binds the intravesicular (surface-exposed) domain of this molecule (published in our own work; Kamin et al., 2010). The curves were identical. Please see the new Figure S6, panel a.

- 2) We compared the FRAP curves for GFP-tagged VGLUT1 with FRAP curves obtained in cultured mouse hippocampal neurons from knock-in Venus-tagged VGLUT1 (Herzog et al., 2011). The curves were again identical. Please see the new Figure S6, panel b.
- 3) We performed a similar comparison for GFP-tagged alpha-synuclein, with results from knock-in Syn-GFP mice (under minimal expression, mimicking human disease; Spinelli et al., 2014). The estimated FRAP values, calculated according to the respective publication, are very close to the results we obtained. Please see the new Figure S6, panel c.
- 4) We compared the FRAP curves of the proteins that are known to be exceptionally enriched in synaptic vesicles, and are not present at substantial levels in any other synaptic compartment, to FRAP curves of synaptic vesicles, obtained after labeling the vesicles with an FM dye (Shtrahman et al., 2005). The FRAP curves of these proteins (synaptophysin, synaptogyrin, VGLUT1 and SV2) are virtually identical to the FRAP curves of vesicles labeled with FM 1-43. Please see the new Figure S6, panel d. Also, please note that these proteins show a slower and more limited recovery than synaptotagmin 1, which is known to be present at higher levels on the plasma membrane, and must therefore be more mobile.

3) *Fig. S12 has shown that in the synapse, immobile fraction and mobility time constants correlate. Does this mean mobility simply reflects protein interaction with presynaptic machinery?*

We agree that mobility in the synapse is likely to be strongly influenced by interactions with the presynaptic machinery.

To provide more information on this, we have now analyzed the FRAP curves more thoroughly. Please see the new Figures 6, 7 and 8. Overall, the analysis confirmed that most soluble proteins interact, at different levels, with the presynaptic machinery.

Fig. 7 shows that vesicle enrichment results in slower synaptic mobility. If so, is immobile fraction not a more intuitive measure of a protein's involvement in presynaptic dynamics? What more does movement rate of a mobile species tell us? The authors should explain or demonstrate.

Our interpretation is as follows: most of the proteins analyzed here are known to bind to synaptic vesicles or to other structures such as the presynaptic active zone. As these structures are far less mobile than the individual proteins, they slow down the movement of the proteins, and increase their immobile fractions. The immobile fraction is therefore mainly a measure of the association of these proteins to low-mobility structures.

However, this measurement is not a particularly sensitive one, unlike the movement rate measurement. For example, many previous measurements indicate that VAMP2/synaptobrevin has a relatively high surface fraction, of at least 20%, which is presumably mobile, while synaptophysin has a far lower surface fraction, of around 2%. Therefore synaptophysin should appear to be less mobile than VAMP2. This is confirmed by the fact that the immobile fraction of VAMP2 is lower than that of synaptophysin (42% and 72%, respectively). The large difference in their behaviors, known from the literature, is nevertheless not strongly reflected by the immobile fraction. In contrast, the difference is far greater for the movement rates (5.5 seconds and 24.6 seconds, respectively). This suggests that the movement rate is more suited to analyzing protein behaviors than the immobile fraction.

An additional point is that the immobile fraction depends on the experiment time scale more than the movement rate, and is therefore a less precise parameter. The immobile fraction is mainly due to the slow exchange of organelles like the synaptic vesicles. Therefore, longer imaging protocols would result in lower immobile fractions, as they allow for more organelle exchange to take place. Such protocols would have no influence on the initial kinetics of movement, which are registered by the changes in the FRAP curve taking place during the first few tens of seconds.

4) Why does this correlation between immobile fraction and movement rate break down in the axons (Fig. S12)?

Movement in the synapse is dominated by interactions to structures such as synaptic vesicles or other presynaptic components. If a protein spends most of its time bound to synaptic vesicles, both the apparent movement and the immobile fraction will be influenced by the same parameter – the fraction of time bound to vesicles. Therefore, movement and immobile fraction will correlate.

Movement in the axon is probably more representative for the behavior of the proteins in the absence of numerous interactions with synaptic structures. The immobile fraction tends to be smaller, and the movement rate is faster. As the two parameters are no longer dominated by one cause (the interaction to synaptic vesicles), they no longer correlate.

5) The authors mentioned criticism of quantum dot tracking to justify the use of FRAP. The authors should briefly summarize the criticism cited in Lee et al., 2017- and discuss the relative bulk added by the GFP molecule as opposed to a quantum dot of different sizes.

We now present this issue in more detail in the first section of Results.

6) Fig. S1 shows consistently lower synaptic fractions in GFP-tagged experiments compared with immunostaining. Why? How does this affect subsequent data interpretation?

This is most likely an effect produced by immunostaining using conventional antibodies. The strong binding of antibodies to their targets depends on an avidity effect. Their probability to stay bound to targets is higher than for monovalent probes because, when one binding pocket unbinds from the target, the other is still bound, thus strongly increasing the probability that the antibodies remain bound. However, this effect has an important downside. It only takes place in areas where the target is abundant, and both pockets can engage in target binding simultaneously. In areas with lower target densities, as outside synapses, the avidity effect is eliminated, and the antibodies have a much higher chance to be washed away, therefore providing an apparent higher synaptic fraction for the antibodies. We have verified this effect for syntaxin and SNAP25 by immunostaining these proteins using antibodies or with nanobodies, which we have recently generated (Maidorn et al., 2019).

7) The authors test the role for active transport by assessing the dynamics of organelles during their imaging window. The paragraph describing figure 5 does not really describe the figure or the rationale for the experiment clearly. A more direct assessment would be to depolymerize microtubules or actin immediately prior to bleaching- allowing them to extract any role of active transport from the recovery curve itself.

The main purpose of these experiments was to determine whether a major fraction of the movement registered by FRAP, especially for soluble proteins, was due to active transport. This

was not the case, which implies that the interpretation of the respective results can be made by assuming that virtually all molecules diffuse in the axon and in the synapse, rather than being actively transported. This was important for the FRAP interpretation models (the new Figure 6). We have re-organized the manuscript to indicate this more clearly.

An example of a non-organelle associated protein should also be shown in the figure.

We now provide an analysis of cytosolic GFP. Please see the new Figure 5.

8) In Fig. 6, could the synaptic species be less mobile simply because of the geometry, as mentioned in the Discussion? How does one distinguish this effect from vesicle enrichment (Fig. 7)?

The synapse geometry very likely has a strong influence on protein movement.

To answer the comment of the Referee, we have performed a very detailed analysis of the FRAP curves. We modelled particle movement in the environment of the synapse and of the axon, with or without binding to synaptic vesicles.

This analysis indicated that the synapse geometry does affect movement, even for GFP, which is not binding to vesicles (the new Figure 6 f). GFP moves rapidly in the axon, but is slowed in the synapse by about 2-fold (the new Figure 8). The same observation was made for all membrane proteins, even for those that do not enrich in synaptic vesicles: all moved faster in the axon, and became slower in the synapse, simply because of the local geometry.

9) Fig. 7C left correlation looks suspicious. How much does upper right datapoint influence the correlation? Why is it such an outlier?

The Referee is right. The correlation depends on this outlier. This is synapsin, a molecule that is known to be able to form a liquid-liquid phase with synaptic vesicles, and which is therefore both strongly bound to vesicles, and limited in its motion by the vesicle interaction. We have now removed this figure panel.

10) It is not clear why the authors thought to look for correlations at the mRNA level. It is also not clear if the authors are comparing the endogenous mRNA or their FRAP reporter mRNA. The authors should elaborate.

We compared the movement values to the known mRNA levels of the respective endogenous proteins. The comparison was made to test whether protein movement can be linked to the protein sequences. As we mentioned in the Introduction, many functional protein parameters are known to depend on the respective protein and mRNA sequences. Such parameters include the protein abundances, lifetimes and translation rates. We therefore sought to test whether such correlations could also be apparent for parameters that are less evidently bound to the sequences, as the protein mobility. We could expand on this further, but the manuscript size has now increased substantially, by including the new controls and the FRAP analysis work. We therefore did not expand the section dealing with the mRNA correlations. We could do so, if the Referee feels it is necessary.

Referee # 3

Summary of the major findings: In this manuscript the authors use FRAP to infer the mobility of 45 GFP-tagged proteins in cultured hippocampal neuron axons and synapses.

We thank the Referee for the comments.

Major shortcomings:

1. The study appears unfocused. I had trouble pinpointing the precise question that this study was trying to address. Further there is insufficient conceptual advancement in this study and a lack of physiological relevance.

The Referee is right. We did not include a major effort we made to analyze the data further, to obtain realistic protein diffusion rates. We initially thought that our study would then be too large to read. We have now included this. Please see the new Figures 6, 7 and 8, and the respective supplementary figures. Overall, the new work now adds diffusion coefficient estimates for all of the proteins, in the synapse, in the axon and in the synaptic vesicle cluster. These estimates should prove important for future computational modeling works on presynaptic physiology.

We could go on and include a section on modeling presynaptic physiology, based on our data. We have performed this, and we used our measurements to calculate whether exocytosis cofactors are limiting during prolonged activity, or whether the well-known reduction in release under such conditions should be ascribed to other parameters, as the reduction of vesicle delivery to the active zones or the slow clearing of vesicle components from the exocytosis sites. We modeled the potential recruitment of dynamin, clathrin and Hsc70 (as an uncoating cofactor). The results indicated that the amounts of dynamin should not be limiting at normal activity rates, but that an absolute requirement for clathrin in the endocytosis of every vesicle could not be supported. However, we decided not to include these data in our already very large manuscript, as they are beyond its scope as a Resource, and as such data could be produced with higher quality in the future by *bona fide* modeling groups.

Importantly, we now also show a large number of controls and validations for the FRAP work. Please refer to our replies to Referee 1, where the respective experiments are described in detail.

2. They claim previous studies tracking mobility of these proteins did not enable analysis of protein structure in relation to synaptic mobility. Using FRAP does not enable protein structure analysis and FRAP does not provide molecular trajectory analysis. There are other techniques that can be utilized to assess this, which have already been carried out in the literature. Additionally, the authors suggest they are generating a protein mobility database for future modeling analysis and that they look at only basal culture conditions as network stimulation would not be physiologically relevant. However, these results are not the definitive protein mobilities, but ensemble measurements specific only to GFP-overexpressed proteins in hippocampal neuron cultures. This data therefore falls short of their aim of generating a protein mobility database, and does not provide any further biological insights for the individual proteins of interest as many have already been studied which they themselves state.

As the first two referees indicate, “molecular mobility of synaptic proteins has hardly been investigated, at least not in a systematic comparative manner” (cited from Referee 1).

The protein tracking techniques the Referee refers to, and which we have also employed for synaptotagmin 1 (Westphal et al., 2008; Kamin et al., 2010) have not provided sufficient analyses for an overall understanding of synaptic proteins.

At the same time, in our initial manuscript we did stop short of analyzing the data to provide a realistic interpretation of protein movement. We agree with the Referee that this reduced the appeal of our work. We provide this thorough analysis now, and we hope that this now makes the novelty of our work clearer.

3. Figure 1 should provide references for each protein to accompany the validation score.

All references are shown in Table S1. They could also be added directly to the figure, if the Referee feels this is necessary.

4. Figure 2a does not sufficiently show that tagged proteins localize to synaptic and axonal areas. Several labels should be used to identify these compartments, and the localization of GFP in relation to these structures.

This figure only shows the overall morphology of an individual neuron. An analysis of expression in synaptic boutons can be performed by investigating the GFP location in relation to synaptophysin immunostainings. This is shown in Figure 3.

5. I have serious concerns over the over-expression analysis. They quote in the introduction they only investigated neurons with mild expression of the GFP-tagged constructs and in the figure 3 legend with moderate expression levels. What they do they suggest is mild/moderate? What is the exact threshold value used to determine whether the neuron could be analysed?

The assessment of overexpression levels was made for each experiment by a brief visual investigation of all neurons in the respective cultures. This was followed by the FRAP experiments, in which only neurons with moderate expression were targeted.

In the materials and methods for the overexpression analysis it is not clear what they define as 'non-overexpressing boutons'. Is this an Immunolabelling of endogenous (non-infected) neurons to establish the threshold? Or have they assumed a threshold for overexpression within the infected neuron population? In which case how have they come to this threshold value to separate these 2 populations?

For Figure 3 random neurons were imaged after the immunostaining, and their expression levels were then analyzed. Unlike in the FRAP experiments, here we analyzed all transfected neurons on the respective coverslips, irrespective of transfection levels. We used the synaptophysin immunostaining channel to determine the positions of the boutons, and we then analyzed the GFP signals in the respective boutons. The boutons with GFP signals significantly higher than the background were considered to belong to transfected neurons. This was confirmed by a visual investigation of the resulting regions of interest. As the transfection procedures we used result in low numbers of transfected neurons, the number of non-transfected neurons and boutons was substantially higher than the number of transfected ones, in all images analyzed.

This method was used in an extensive work we published in the same journal (Truckenbrodt et al., 2018), using similar cultures.

6. In figure 3 they state the error bar indicates the 75th percentile. This suggests the

overexpression levels have a greater range than those visible in figure 3, and thus the overexpression levels do not just range from 1.2-2 fold as stated in the introduction.

In the Introduction we referred to the average overexpression levels. We now changed this statement.

7. I would suggest they consult a statistician. The use of the Benjamini-Hochberg procedure in figure 4 is not fully justified, as these are strictly only bivariate comparisons of the overexpression level vs mobility in each individual protein. The n number is also relatively low and thus the use of the Benjamini-Hochberg procedure can easily induce false negatives in such small populations. Multiple tests are not being carried out on the same protein, and each protein is independent of the others, thus they are not correcting for multiple testing.

We agree with the Referee that “*thus they are not correcting for multiple testing*”. The Benjamini-Hochberg analysis was preferred precisely because it does not correct strongly for multiple testing, unlike a Bonferroni procedure, and thus works against us, allowing us to detect all possible correlations.

At any rate, very few correlations were significant, even before thinking of correcting for multiple testing. The P values we reported were not corrected for multiple testing, and therefore one can easily evaluate the individual correlations.

8. It would be more helpful to see the linear regression plotted on each of the graphs shown in figure 4.

We now provide this as a large supplement, for all proteins. Please see the new Figure S5.

9. What is the spot size of the FRAP area? And what is the area being used to analyse recovery rates? These values will have a huge influence on the time constants.

The spot size of the FRAP area was a disk with a radius of 1 μm . The recovery rates were analyzed in this area. The frame size was of 3.23 x 3.23 μm^2 .

The new analysis we provide (please see Figure 6, 7 and 8) took these values into account, to estimate realistic movement characteristics for our proteins.

10. The tracking analysis of organelle-bound proteins cannot be extrapolated to deduce whether active organelle processes are affecting the FRAP estimates for all the analysed proteins.

The main purpose of these experiments was to determine whether a major fraction of the movement registered by FRAP, especially for soluble proteins, was due to active transport. This was not the case, which implies that the interpretation of the respective results can be made by assuming that virtually all molecules diffuse in the axon and in the synapse, rather than being actively transported. This was important for the FRAP interpretation models (the new Figure 6). We have re-organized the manuscript to indicate this more clearly, and we have removed most of the unnecessary details.

11. The materials and methods section on the organelle tracking analysis is not detailed enough to understand how the images were acquired or how the analysis was carried out.

The explanation for the imaging procedure was presented in the Methods in the section termed “Organelle imaging for tracking purposes”, which was presented before the section “Organelle tracking analysis”. We have now combined these sections under the same heading, “Organelle tracking analysis”.

We have now explained the analysis in more detail. We also provide an address for a download of the codes: <http://site.physics.georgetown.edu/matlab/code.html>

12. The authors make an assumption that the mobility they observe accounts for protein-specific interactions and that this confirms their expression conditions, yet the data does not directly support this claim. The wording is very misleading as they also state they compare FRAP time constraints with strength of association of each protein to synaptic vesicles, yet they do not measure association strength.

We have performed a detailed analysis of protein motion that adds substantial information to this issue. We hope that the new data are more informative than our older figures. Please see the new Figures 6, 7 and 8, and the respective supplementary data and Methods sections.

13. The authors state the fold difference between synaptic and axonal time constants is similar to the fold difference between membrane and soluble proteins and that this suggests the synaptic environment has a similar influence on the movement of both types of proteins. Only absolute values of time constants could be used to deduce this, not fold changes.

We have removed the respective panel.

14. In the discussion the authors compare FRAP only to FCS and not single particle tracking methods and thus a global view of the literature is missing.

This comparison was not made in the discussion, but rather in the Results, due to the fact that we actually employed FCS in our experiments. We have now added a paragraph discussing particle tracking to the respective section of Results.

15. Figure S12 - why do they have negative immobile fractions?

Negative immobile fractions appear mostly for proteins that are highly mobile, and where the entry of proteins into the bleached spot reaches values above the initial intensity. We analyze small areas within synapses or axons, rather than large cellular areas where the entry and exit of proteins are expected to balance each other perfectly. Therefore one has to expect that on some measurements the protein entry into the bleached spots surpasses the initial fluorescence levels.

16. Grammatical errors and misleading text throughout.

We have thoroughly checked our manuscript, and we hope we have been able to eliminate all of these problems.

References

van Bommel, B., Konietzny, A., Kobler, O., Bär, J., and Mikhaylova, M. (2019). F-actin patches associated with glutamatergic synapses control positioning of dendritic lysosomes. *EMBO J.* 38, e101183.

- Herzog, E., Nadrigny, F., Silm, K., Biesemann, C., Helling, I., Bersot, T., Steffens, H., Schwartzmann, R., Nägerl, U.V., El Mestikawy, S., et al. (2011). In vivo imaging of intersynaptic vesicle exchange using VGLUT1 Venus knock-in mice. *J. Neurosci. Off. J. Soc. Neurosci.* *31*, 15544–15559.
- Kamin, D., Lauterbach, M.A., Westphal, V., Keller, J., Schönle, A., Hell, S.W., and Rizzoli, S.O. (2010). High- and Low-Mobility Stages in the Synaptic Vesicle Cycle. *Biophys. J.* *99*, 675–684.
- Koskinen, M., and Hotulainen, P. (2014). Measuring F-actin properties in dendritic spines. *Front. Neuroanat.* *8*.
- Koskinen, M., Bertling, E., and Hotulainen, P. (2012). Methods to measure actin treadmilling rate in dendritic spines. *Methods Enzymol.* *505*, 47–58.
- Petersen, A., and Gerges, N.Z. (2015). Neurogranin regulates CaM dynamics at dendritic spines. *Sci. Rep.* *5*, 1–10.
- Shtrahman, M., Yeung, C., Nauen, D.W., Bi, G., and Wu, X. (2005). Probing Vesicle Dynamics in Single Hippocampal Synapses. *Biophys. J.* *89*, 3615–3627.
- Spinelli, K.J., Taylor, J.K., Osterberg, V.R., Churchill, M.J., Pollock, E., Moore, C., Meshul, C.K., and Unni, V.K. (2014). Presynaptic alpha-synuclein aggregation in a mouse model of Parkinson's disease. *J. Neurosci. Off. J. Soc. Neurosci.* *34*, 2037–2050.
- Takamori, S., Holt, M., Stenius, K., Lemke, E.A., Grønborg, M., Riedel, D., Urlaub, H., Schenck, S., Brügger, B., Ringler, P., et al. (2006). Molecular Anatomy of a Trafficking Organelle. *Cell* *127*, 831–846.
- Truckenbrodt, S., Viplav, A., Jähne, S., Vogts, A., Denker, A., Wildhagen, H., Fornasiero, E.F., and Rizzoli, S.O. (2018). Newly produced synaptic vesicle proteins are preferentially used in synaptic transmission. *EMBO J.* e98044.
- Westphal, V., Rizzoli, S.O., Lauterbach, M.A., Kamin, D., Jahn, R., and Hell, S.W. (2008). Video-rate far-field optical nanoscopy dissects synaptic vesicle movement. *Science* *320*, 246–249.

Dear Silvio,

Thank you for submitting your revised manuscript to the EMB Journal. Your study has now been seen by the original referees #1 and 2 and their comments are provided below. As you can see both of them support publication here.

I am therefore very happy to let you know that we will publish your manuscript here. Before sending you the formal acceptance letter there are just a few issues that need to be resolved.

- I was trying to see if there was a clever way to deal with the supplemental figures as there are quite a few of them, but I couldn't think of a better way to deal with it so OK as is. The supplemental files need to be added together in one appendix file with a ToC. Please add the figure legends for the supplemental figures in this file as well.

- For any panels in the main figures reused in the supplemental figures - please add a note to this in the figure legend for example panels: Fig 2EF/S4, Fig 2B,D/S2 and Fig 6A/S17A

- Are the references in the table S1 also provided in the references in the main MS file

- Please remove the figures from the MS file and add the figure legends after the references

- Can you please check for figure callouts in the main MS text for the following panels: Fig 3 panels, Fig 4C panel, Fig 5a, Fig 6A, C-E, G-H

- We need 5 keywords

- Please re-label competing interest as Conflict of Interest

- Author contribution is missing Eugenio Fornaseiro

- Please Zip each movie with its figure legend

- We require a Data Availability Section. As far as I can see the manuscript doesn't contain data that is deposited in external databases. If so then add: This study includes no data deposited in external repositories or something similar.

- Our publisher has also done their pre-publication check on your manuscript. When you log into the manuscript submission system you will see the file "Data edited manuscript file". Please take a look at the word file and the comments regarding the figure legends and respond to the issues. Please also use this version when you resubmit the revised version with the marked changes. Just makes it easier for us to see the changes.

- We encourage the publication of source data, particularly for electrophoretic gels and blots, with the aim of making primary data more accessible and transparent to the reader. It would be great if you could provide me with a PDF file per figure that contains the original, uncropped and unprocessed scans of all or key gels used in the figure? The PDF files should be labeled with the appropriate figure/panel number, and should have molecular weight markers; further annotation could be useful but is not essential. The PDF files will be published online with the article as

supplementary "Source Data" files.

- We include a synopsis of the paper (see <http://emboj.embopress.org/>). Please provide me with a general summary statement and 3-5 bullet points that capture the key findings of the paper.

That should be all! Let me know if we need to discuss anything further

With best wishes

Karin

Karin Dumstrei, PhD
Senior Editor
The EMBO Journal

Further information is available in our Guide For Authors:

The revision must be submitted online within 90 days; please click on the link below to submit the revision online before 5th Aug 2020.

Link Not Available

Referee #1:

The authors went through efforts of responding my critique. I can largely follow their arguing, and now recommend publication of the manuscript.

Referee #2:

The authors have mostly addressed my previous concerns and suggestions and I now find it fine for publication.

Dear Silvio,

Thank you for submitting your revised manuscript to The EMBO Journal. I have now had a chance to take a look at it and I am happy with the introduced changes.

I am therefore very pleased to accept the manuscript for publication here.

just two minor things => In the legend to Fig 6H it refers to "supplemental movies". We should fix the call out to EV movies. Can you send me a modified text file via email and then we will replace it for you.

We also need a text synopsis => a summary statement and 3-5 bullet points.

That is all - Congratulations on a nice study

With best wishes

Karin

Karin Dumstrei, PhD
Senior Editor
The EMBO Journal

Please note that it is EMBO Journal policy for the transcript of the editorial process (containing referee reports and your response letter) to be published as an online supplement to each paper. If you do NOT want this, you will need to inform the Editorial Office via email immediately. More information is available here: http://emboj.embopress.org/about#Transparent_Process

Your manuscript will be processed for publication in the journal by EMBO Press. Manuscripts in the PDF and electronic editions of The EMBO Journal will be copy edited, and you will be provided with page proofs prior to publication. Please note that supplementary information is not included in the proofs.

Should you be planning a Press Release on your article, please get in contact with embojournal@wiley.com as early as possible, in order to coordinate publication and release dates.

If you have any questions, please do not hesitate to call or email the Editorial Office. Thank you for your contribution to The EMBO Journal.

** Click here to be directed to your login page: <http://emboj.msubmit.net>

Corresponding Author Name: Silvio Rizzoli

Manuscript Number: EMBOJ-2020-104596R